# TEG-DB: A Comprehensive Dataset and Benchmark of Textual-Edge Graphs

**Zhuofeng Li**[*△]  **Zixing Gou**[*▽]  **Xiangnan Zhang**[◇]  **Zhongyuan Liu**[°]  **Sirui Li**[†]
**Yuntong Hu**[†]  **Chen Ling**[†]  **Zheng Zhang**[†]  **Liang Zhao**[†]
[△]Shanghai University  [▽]Shandong University  [◇]Johns Hopkins University  [°]China
University of Petroleum (East China)  [†]Emory University

## Abstract

Text-Attributed Graphs (TAGs) augment graph structures with natural language descriptions, facilitating detailed depictions of data and their interconnections across various real-world settings. However, existing TAG datasets predominantly feature textual information only at the nodes, with edges typically represented by mere binary or categorical attributes. This lack of rich textual edge annotations significantly limits the exploration of contextual relationships between entities, hindering deeper insights into graph-structured data. To address this gap, we introduce Textual-Edge Graphs Datasets and Benchmark (TEG-DB), a comprehensive and diverse collection of benchmark textual-edge datasets featuring rich textual descriptions on nodes and edges. The TEG-DB datasets are large-scale and encompass a wide range of domains, from citation networks to social networks. In addition, we conduct extensive benchmark experiments on TEG-DB to assess the extent to which current techniques, including pre-trained language models (PLMs), graph neural networks (GNNs), proposed novel entangled GNNs and their combinations, can utilize textual node and edge information. Our goal is to elicit advancements in textual-edge graph research, specifically in developing methodologies that exploit rich textual node and edge descriptions to enhance graph analysis and provide deeper insights into complex real-world networks. The entire TEG-DB project is publicly accessible as an open-source repository on Github, accessible at https://github.com/Zhuofeng-Li/TEG-Benchmark.

## 1 Introduction

Text-attributed graphs (TAGs) are graph structures in which nodes are equipped with rich textual information, allowing for deeper analysis and interpretation of complex relationships [50, 18, 16]. TAGs are widely utilized in a variety of real-world applications, including social networks [33, 32], citation networks [26], and recommendation systems [44, 15]. Due to the universal representational capabilities of language, TAGs have emerged as a promising format for potentially unifying a wide range of existing graph datasets. This field has recently garnered rapidly growing interest, particularly in the development of foundational models for graph data [24, 16, 46].

Unfortunately, a central issue in designing the TAG foundation model is the lack of comprehensive datasets with rich textual information on both nodes and edges. Most traditional graph datasets solely offer node attribute embeddings, devoid of the original textual sentences, which results in a significant loss of context and limits the application of advanced techniques such as large language models (LLMs) [25]. Despite some TAG datasets being present recently [46], their data usually only have text information on nodes where the edges are usually represented as binary or categorical. However, the textual information of edges in TAGs is crucial for elucidating the meaning of individual documents and their semantic correlations. For instance, as shown in Figure 1, this scientific article network illustrates the citation patterns of articles authored by Einstein and Planck in the field of quantum mechanics. When we need to conclude that 'Planck endorsed the probabilistic nature of quantum mechanics while Einstein opposed this view,' and if we consider it in terms of a TAG view, focusing

---

38th Conference on Neural Information Processing Systems (NeurIPS 2024) Track on Datasets and Benchmarks.

 * These two authors contribute equally to this paper.

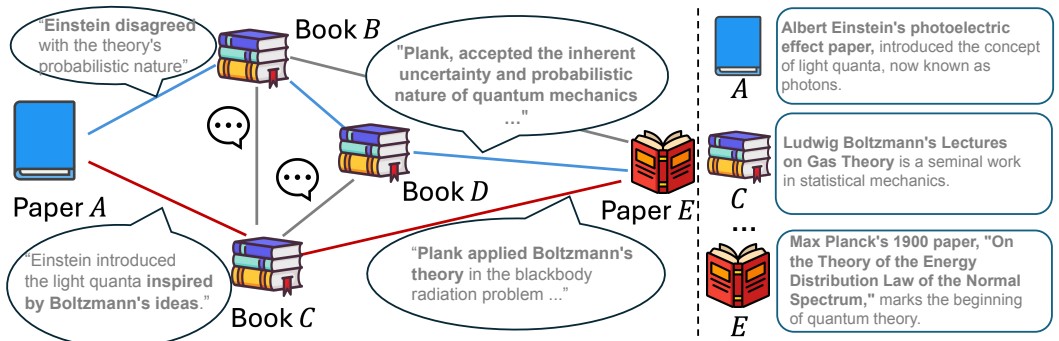

Figure 1: An example of textual-edge graph about scientific article network in quantum theory: two papers are connected by citation links. Considering edge texts in the TEG enhances semantic understanding and improves text analysis.

solely on the content of the papers authored by Einstein (Paper A) and Planck (Paper E), we would only conclude that both Einstein and Planck supported quantum mechanics. However, to further deduce that Einstein opposed studying quantum mechanics from a probabilistic perspective, it is necessary to adopt the Textual-Edge Graph (TEG) approach. This approach not only focuses on the paper contents but also pays greater attention to the citation information from the edge between Paper A and Book B, as well as the edge between Paper E and Book D. These edges provide essential citation context and reveal the relationships and influence between different scholarly works.

While compelling, TEGs face three significant challenges that make them an open problem. (1) *Comprehensive TEG datasets are absent.* Currently, there is a lack of comprehensive TEG datasets that simultaneously incorporate textual information from both nodes and edges, spanning multiple domains of varying sizes, and encompassing various mainstream graph learning tasks. This deficiency hinders the evaluation of TEG-based methods across diverse applications and domains. (2) *Existing experimental settings for TEG are disorganized.* Due to the inherent variety and complexity of TEGs, coupled with the absence of a standardized data format, existing works have adopted different datasets with different experimental settings [19, 18, 17, 53, 52, 23, 24]. This causes great difficulties in model comparisons in this field. (3) *Comprehensive benchmarks and analyses for TEG-based methods are missing.* While some techniques can accommodate edge features, they typically process binary or categorical data. It remains unclear if these methods can effectively utilize rich textual information on edges, particularly in leveraging complex interactions between graph nodes.

**Present work.** Recognizing all the above challenges, our research proposes the Textual-Edge Graphs Datasets and Benchmark (TEG-DB). TEG-DB is a pioneering initiative offering a diverse collection of benchmark graph datasets with rich textual descriptions on both nodes and edges. To address the issue of inadequate TEG datasets, our TEG datasets as shown in Table 1 cover an extensive array of domains, including Book Recommendation, E-commerce, Academic, and Social networks. Ranging in size from small to large, each dataset contains abundant raw text data associated with both nodes and edges, facilitating comprehensive analysis and modeling across various fields. Moreover, to address the inconsistency in experimental settings and the lack of comprehensive analyses for TEG-based methods, we first represent the TEG dataset in a unified format, then conduct extensive benchmark experiments and perform a comprehensive analysis. These experiments are designed to evaluate the capabilities of current computational techniques, such as pre-trained language models (PLMs), graph neural networks (GNNs) and proposed novel entangled GNNs, as well as their integrations. Our contributions are summarized below:

- To the best of our knowledge, TEG-DB is the first open dataset and benchmark specifically designed for textual-edge graphs. We provide 9 comprehensive TEG datasets encompassing 4 diverse domains as shown in Table 1. Each dataset, varying in size from small to large, contains abundant raw text data associated with both nodes and edges. Our TEG datasets aim to bridge the gap of TEG dataset scarcity and provide a rich resource for advancing research in the TEG domain.

- We develop a standardized pipeline for TEG research, encompassing crucial stages such as data preprocessing, data loading, and model evaluation. With this framework, researchers can seamlessly replicate experiments, validate findings, and iterate on existing approaches with greater efficiency

and confidence. Additionally, this standardized pipeline facilitates collaboration and knowledge sharing within the TEG community, fostering innovation and advancement in the field.

- We conduct extensive benchmark experiments and perform a comprehensive analysis of TEG-based methods, delving deep into various aspects such as the impact of different models and embeddings generated by PLMs of various scales, the consequence of diverse embedding methods in GNNs including separate and entangled embeddings, the effect of edge text and the influence of different domain datasets. By addressing key challenges and highlighting promising opportunities, our research stimulates and guides future directions for TEG exploration and development.

## 2  Related Works

In this section, we will begin by providing a brief introduction to three commonly used learning paradigms for TAGs. Following this, we will delve into the comparisons between the current graph learning benchmarks and our proposed benchmark.

**PLM-based methods.** PLM-based methods leverage the power of PLM to enhance the text modeling within each node due to their pre-training on a vast corpus. The early works on modeling textual attributes were based on shallow networks, e.g., Skip-Gram [30] and GloVe [34]. In recent years, Large Language Models (LLM) have become trending tools. Models like Llama [38], PaLM [2], and GPT [1] show their strong comprehension and inferring ability in cross-field natural language based tasks like code generation [3], legal consulting [6], make creative arts [22], as well as understanding and learning from Graphs [5]. One of the key applications of pre-trained language models is text representation, in which low-dimensional embeddings capture the underlying semantics of texts. On the TAGs, the PLMs use the local textual information of each node to learn a good representation for the downstream task.

**GNN-based methods**. The rapid advancements in graph representation learning within machine learning have led to numerous studies addressing various tasks, such as node classification [21] and link prediction [51]. Graph neural networks (GNNs) are acknowledged as robust tools for modeling graph data. These methods, including GCN [21], GAT [39], GraphSAGE [10], GIN [45], and RevGAT [31], develop effective message-passing mechanisms that facilitate information aggregation between nodes, thereby enhancing graph representations. GNNs typically utilize the "cascade architecture" advocated by GraphSAGE for textual graph representation, wherein node features are initially encoded independently using text modeling tools (e.g., PLMs) and then aggregated by GNNs to generate the final representation.

**LLM as Predictor.** In recent years, several recent studies [47, 4, 9] have delved into the potential of Large Language Models (LLMs) in analyzing graph-structured data. However, there is a lack of comprehensive research on the ability of LLMs to effectively identify and utilize key topological structures across various prompt scenarios, task complexities, and datasets. Chen et al. [4] and Guo et al. [9] proposed using LLMs on graph data but primarily focused on node classification within specific citation network datasets, limiting the exploration of LLMs' performance across various tasks and datasets. Furthermore, Ye et al. [47] fine-tuned LLMs on a specific dataset to outperform GNNs, focusing on a different research goal, which emphasizes LLMs' inherent ability to understand and leverage graph structures.

**Benchmarks for text-attributed graphs.** Current benchmarks in text-attributed graph representation learning can be divided into two stages. The first stage benchmark includes datasets such as mag [42] and ogbn-arxiv [13], which feature limited textual information primarily associated with nodes. The second stage benchmark is represented by CS-TAG [46], which builds upon the first stage by providing richer node-level textual data. However, these datasets face limitations in exploring representation learning for textual-edge graphs. Specifically, they typically include text only on nodes, with edges often represented as binary or categorical, which restricts a comprehensive understanding of node semantic relationships. Additionally, they lack coverage across diverse domains and tasks, hindering the development of robust and generalizable models. Furthermore, the lack of uniformity in representation formats introduces inconsistencies and complexities in analysis and modeling. Thus, there is a clear need for the development of a comprehensive benchmark with textual information on both nodes and edges in a unified format.

Table 1: Comparison between our TEG-DB datasets and existing datasets on TAG.

| | Dataset | Nodes | Edges | Nodes-Class | Graph Domain | Size | Nodes-text | Edges-text | Node Classification | Link Prediction |
|---|---|---|---|---|---|---|---|---|---|---|
| Previous | Twitch Social Network [35] | 7,126 | 88,617 | 2 | Social Networks | Small | ✗ | ✗ | ✓ | ✗ |
| | Facebook Page-Page Network [36] | 22,470 | 171,002 | 4 | Social Networks | Small | ✗ | ✗ | ✓ | ✗ |
| | ogbn-arxiv [13] | 169,343 | 1,166,243 | 40 | Academic | Medium | ✓ | ✗ | ✓ | ✗ |
| | Citeseer [37] | 3,327 | 4,732 | 6 | Academic | Small | ✗ | ✗ | ✓ | ✗ |
| | Pubmed [37] | 19,717 | 44,338 | 3 | Academic | Small | ✗ | ✗ | ✓ | ✗ |
| | Cora [28] | 2,708 | 5,429 | 7 | Academic | Small | ✗ | ✗ | ✓ | ✗ |
| | CitationV8 [46] | 1,106,759 | 6,120,897 | - | Academic | Large | ✓ | ✗ | ✗ | ✓ |
| | GoodReads [46] | 676,084 | 8,582,324 | 11 | Book Recommendation | Large | ✓ | ✗ | ✗ | ✓ |
| | Sports-Fitness [46] | 173,055 | 1,773,500 | 13 | E-commerce | Medium | ✓ | ✗ | ✓ | ✗ |
| | Ele-Photo [46] | 48,362 | 500,928 | 12 | E-commerce | Small | ✓ | ✗ | ✓ | ✗ |
| | Books-History [46] | 41,551 | 358,574 | 12 | E-commerce | Small | ✓ | ✗ | ✓ | ✗ |
| | Books-Children [46] | 76,875 | 1,554,578 | 24 | E-commerce | Small | ✓ | ✗ | ✓ | ✗ |
| | ogbn-arxiv-TA [46] | 169,343 | 1,166,243 | 40 | Academic | Medium | ✓ | ✗ | ✓ | ✗ |
| Ours | Goodreads-History | 540,807 | 2,368,539 | 11 | Book Recommendation | Large | ✓ | ✓ | ✓ | ✓ |
| | Goodreads-Crime | 422,653 | 2,068,223 | 11 | Book Recommendation | Large | ✓ | ✓ | ✓ | ✓ |
| | Goodreads-Children | 216,624 | 858,586 | 11 | Book Recommendation | Large | ✓ | ✓ | ✓ | ✓ |
| | Goodreads-Comics | 148,669 | 631,649 | 11 | Book Recommendation | Medium | ✓ | ✓ | ✓ | ✓ |
| | Amazon-Movie | 137,411 | 2,724,028 | 399 | E-commerce | Medium | ✓ | ✓ | ✓ | ✓ |
| | Amazon-Apps | 31,949 | 62,036 | 62 | E-commerce | Small | ✓ | ✓ | ✓ | ✓ |
| | Reddit | 478,022 | 676,684 | 3 | Social Networks | Large | ✓ | ✓ | ✓ | ✓ |
| | Twitter | 18,761 | 23,764 | 505 | Social Networks | Small | ✓ | ✓ | ✓ | ✓ |
| | Citation | 169,343 | 1,166,243 | 40 | Academic | Large | ✓ | ✓ | ✓ | ✓ |

# 3 Preliminaries

A Textual-Edge Graph (TEG) is a graph-structured data format in which both nodes and edges have free-form text descriptions. These textual annotations provide rich contextual information about the complex relationships between entities, enabling a more detailed and comprehensive representation of data relations than traditional graphs.

**Definition 1** (Textual-edge Graphs). Formally, a TEG can be represented as $\mathcal{G} = (\mathcal{V}, \mathcal{E})$, which consists of a set of nodes $\mathcal{V}$ and a set of edges $\mathcal{E} \subseteq \mathcal{V} \times \mathcal{V}$. Each node $v_i \in \mathcal{V}$ contains a textual description $d_i$, and each edge $e_{ij} \in \mathcal{E}$ also associates with its text description $d_{ij}$ describing the relation between $v_i$ and $v_j$.

**Challenges.** Current research on TEGs faces three significant challenges: (1) The scarcity of large-scale, diverse TEG datasets; (2) Inconsistent experimental setups and methodologies in previous TEG research; and (3) The absence of standardized benchmarks and comprehensive analyses for evaluating TEG-based methods. These limitations impede the development of more effective and efficient approaches in this emerging field.

# 4 A Comprehensive Dataset and Benchmark of Textual-Edge Graphs

We begin by offering a brief overview of the TEG-DB in Section 4.1. Afterward, we provide a comprehensive overview of the TEG datasets in Section 4.2, detailing their composition and the preprocessing steps to represent them in a unified format. Finally, we discuss three main methods for handling TEGs: PLM-based, GNN-based paradigm, and LLM as Predictor methods in Section 4.3.

## 4.1 Overview of TEG-DB

In order to overcome the constraints intrinsic to preceding studies, we propose the establishment of the Textual-Edge Graphs Datasets and Benchmark, referred to as TEG-DB. This framework functions as a standardized evaluation methodology for examining the effectiveness of representation learning approaches in the context of TEGs. To ensure the comprehensiveness and scalability of TEG datasets, TEG-DB collects and constructs a novel set of datasets covering diverse domains like book recommendation, e-commerce, academia, and social networks, varying in size from small to large. These datasets are suitable for various mainstream graph learning tasks such as node classification and link prediction. Table 1 compares previous datasets with our TEG datasets. To enhance usability, we unify the TEG data format and propose a modular pipeline with three main methods for handling TEGs. To further foster TEG model design, we extensively benchmark TEG-based methods and conduct a thorough analysis. Overall, TEG-DB provides a scalable, unified, modular, and regularly updated evaluation framework for assessing representation learning methods on textual graphs.

## 4.2 Data Preparation and Construction

In order to construct the dataset with simultaneous satisfaction of both rich textual information on nodes and edges, nine datasets from diverse domains and different scales are chosen. Specifically, we collect four User-Book Review networks from Goodreads datasets [40, 41] in the Book Recommendation domain and two shopping networks from Amazon datasets [11, 12] in the E-commerce

domain. Two social networks from Reddit and Twitter [29]. One citation network from MAG [42] and The Semantic Scholar Open Data Platform [20] in the academic domain. The statistics of the datasets are shown in Table 1.

The creation of textual-edge graph datasets involves three main steps. Firstly, preprocessing the textual attributes within the original dataset, which includes tasks such as handling missing values, filtering out non-English statements, removing anomalous symbols, truncating excessive length and selecting the most relevant textual attributes as the raw text for nodes or edges. Secondly, constructing the TEG itself. The connectivity between nodes is derived from inherent relationships provided within the dataset, such as citation relationships between papers in citation networks. It is important to note that during graph construction, self-edges and isolated nodes are eliminated. Lastly, refining the constructed graph. It is noteworthy that our dataset encompasses all major tasks in graph representation learning: node classification and link prediction. Below are the specifics of each dataset:

**User-Book Review Network.** Four datasets within the realm of User-Book Review Networks, specifically labeled as Goodreads-History, Goodreads-Crime, Goodreads-Children, and Goodreads-Cosmics, were formulated. The Goodreads datasets are the main source. Nodes represent different types of books and reviewers, while edges indicate book reviews. Node labels are assigned based on the book categories. The descriptions of books are used as book node textual information while user information serves as the user node textual information and reviews of users are used as edges textual information. The corresponding tasks are to predict the categories of the books, which is formulated as a multi-label classification problem, and to predict whether there are connections between users and books. These comprehensive data help infer user preferences and identify similar tastes, enhancing online book recommendations, unlike existing datasets that often lack interaction texts.

**Shopping Networks.** Two datasets, Amazon-Apps and Amazon-Movie, are classified under Shopping Networks. The Amazon datasets are the primary source, encompassing item reviews and descriptions. Nodes represent different types of items and reviewers, while edges indicate item reviews. The descriptions of items are used as item node textual information, while user information serves as the user node textual information and reviews of users are used as edge textual information. The corresponding tasks are to predict the categories of the items, formulated as a multi-label classification problem, and to predict whether there are connections between users and items. These datasets have the potential to significantly enhance recommendation systems, providing richer data for more accurate suggestions and a personalized shopping experience.

**Citation Networks.** The raw data for the citation network is sourced from the MAG and The Semantic Scholar Open Data Platform. Nodes represent papers, and edges represent the citation relationship. The titles and abstracts of papers are used as node textual information, and citation information, such as the context and paragraphs in which papers are cited, is utilized as textual edge data. The corresponding task involves predicting the domain to which a paper belongs, formulated as a multi-class classification problem, and predicting whether there exists a citation relationship between papers. This dataset enhances academic network expressiveness, particularly benefiting tasks like node classification and link prediction in graph machine learning.

**Social Networks.** The Reddit dataset, sourced from Reddit and the Twitter dataset, derived from Twitter, represent two prominent social media platforms. Nodes represent users and topics. The edges indicate the post-relationship. The descriptions of topics are used as topic node textual information while user information serves as the user node textual information and post text in subreddits or tweets is used as edge textual information. The corresponding tasks are to predict the category of the topics, formulated as a multi-class classification problem, and to predict whether there are connections between users and topics. Utilizing these datasets enhances recommendation algorithm performance, providing more personalized and relevant suggestions, while also offering valuable insights into user interests and preferences for social network research and business decision-making.

### 4.3   Adapting Existing Methods to Solve Problems in TEGs

**PLM-based Paradigm.** PLMs are trained on massive amounts of text data, allowing them to learn the semantic relationships between words, phrases, and sentences. This enables them to understand the meaning behind the text, not just on a superficial level, but also in terms of context and intent. So PLM-based methods leverage the power of PLM to enhance the text modeling within each node and

edge, along with an extra multilayer perception (MLP) to integrate their textual information from TEG. The formulation of these methods is as follows:

$$\boldsymbol{h}_u^{(k+1)} = \text{MLP}_{\boldsymbol{\psi}}^{(k)}\left(\boldsymbol{h}_u^{(k)}\right)$$
$$\boldsymbol{h}_u^{(0)} = \text{PLM}(T_u) + \sum_{v \in \mathcal{N}(u)} \text{PLM}(T_{e_{v,u}}) \tag{1}$$

where $\boldsymbol{h}_u^{(k)}$ denotes the node representation of node $u$ in layer $k$ of Multilayer Perceptron (MLP). $T_u$ and $T_{e_{v,u}}$ represent the raw text on node $u$ and edge $e_{v,u}$, respectively. The initial feature vector $\boldsymbol{h}_u^{(0)}$ of node $u$ is derived by encoding the text on node $u$ and its neighboring edges using the Pre-trained Language Model (PLM). $\mathcal{N}$ denotes the set of neighbors. $\psi$ refers to the trainable parameters within the MLP.

Although PLMs have considerably improved the representation of node text attributes, these models do not account for topological structures. This limitation hinders their ability to fully capture the complete topological information present in TEGs.

**Edge-aware GNN-based Paradigm.** GNNs are employed to propagate information across the graph, allowing for the extraction of meaningful representations via message passing, which are formally defined as follows:

$$\boldsymbol{h}_u^{(k+1)} = \text{UPDATE}_{\boldsymbol{\omega}}^{(k)}\left(\boldsymbol{h}_u^{(k)}, \text{AGGREGATE}_{\boldsymbol{\omega}}^{(k)}\left(\left\{\boldsymbol{h}_v^{(k)}, \boldsymbol{e}_{v,u}, v \in \mathcal{N}(u)\right\}\right)\right) \tag{2}$$

where $\boldsymbol{h}_u^{(k)}$ denotes the node representation of node $u$ in layer $k$ of GNN and the initial node feature vector $\boldsymbol{h}_u^{(0)}$ is obtained by embedding its raw text through PLMs. $e_{v,u}$ denotes the edge from node $v$ to node $u$ and its features $\boldsymbol{e}_{v,u}$ are likewise derived from PLMs based on its raw text embeddings. $k$ represents the layers of GNNs, $\mathcal{N}$ denotes the set of neighbors, $u$ denotes the target node, $\boldsymbol{\omega}$ means the learning parameters in GNNs.

However, this approach presents two primary issues: (1) Existing Graph ML methods like GNNs typically work on structured attributes on edges instead of texts [18]. In TEGs, edges are texts that contain rich semantic information, which is way beyond the typical focus of GNNs that are commonly based on connectivity (i.e., binary attribute denoting whether there is a connection or not) and edge attributes (i.e., categorical or numerical values on the edges). (2) GNN-based methods are limited in capturing the contextualized semantics of edge texts [46]. In TEGs, where edge and node texts are often entangled, converting them into separate node and edge embeddings during the embedding process can result in the loss of critical information about their interdependence, which diminishes the effectiveness of GNNs throughout the entire message-passing process.

**Entangled GNN-based Paradigm.** Traditional edge-aware GNN-based approaches that first learn edge text embeddings and then apply GNNs have limitations for TEG data because edge texts and node texts are often closely entangled. Separating them into distinct node and edge embeddings may impair important information regarding their interaction. For instance, in a citation graph where each node represents a paper, an edge might indicate that one paper cites, criticizes, or utilizes a specific part of another paper. Therefore, the edge does not represent the relationship between the entirety of the two nodes, posing a significant challenge for methods that rely on node or edge embeddings representing the entirety of a node or edge. To avoid information loss during the interaction between nodes and edges after text embedding, we propose an approach that first entangles the edge text and node text before performing the embedding. The embedding obtained in this way is then added to the message-passing operation for each pair of connected nodes. The formulation of these methods is as follows:

$$\boldsymbol{h}_u^{(k+1)} = \text{UPDATE}_{\boldsymbol{\omega}}^{(k)}\left(\boldsymbol{h}_u^{(k)}, \text{AGGREGATE}_{\boldsymbol{\omega}}^{(k)}\left(\left\{\boldsymbol{h}_v^{(k)}, v \in \mathcal{N}(u)\right\}\right)\right)$$
$$\boldsymbol{h}_u^0 = \text{PLM}(T_u, \{T_v, T_{e_{v,u}}, v \in \mathcal{N}(u)\}) \tag{3}$$

where $\boldsymbol{h}_u^{(k)}$ denotes the node representation of node $u$ in layer $k$ of the GNN. $T_v$, $T_u$, and $T_{e_{v,u}}$ represent the raw text on node $v$, node $u$, and the edge from $v$ to $u$, respectively. The initial node feature vector $\boldsymbol{h}_u^{(0)}$ is obtained by embedding the entangled raw text of node $u$ and its neighborhood

Table 2: Link prediction AUC and F1 among PLM-based, GNN-based methods. The best method for each PLM embedding on each dataset is shown in bold.

| Methods | Children | | | | | | | | | | Crime | | | | | | | | | |
|---|---|---|---|---|---|---|---|---|---|---|---|---|---|---|---|---|---|---|---|---|
| | Entangled-GPT | | GPT-3.5-TURBO | | BERT-Large | | BERT | | None | | Entangled-GPT | | GPT-3.5-TURBO | | BERT-Large | | BERT | | None | |
| | AUC | F1 | AUC | F1 | AUC | F1 | AUC | F1 | AUC | F1 | AUC | F1 | AUC | F1 | AUC | F1 | AUC | F1 | AUC | F1 |
| MLP | 0.9146 | 0.8459 | 0.8952 | 0.8198 | 0.8948 | 0.8193 | 0.8947 | 0.8192 | 0.8929 | 0.8181 | 0.9030 | 0.8429 | 0.8911 | 0.8144 | 0.8909 | 0.8145 | 0.8920 | 0.8153 | 0.8913 | 0.8149 |
| GraphSAGE | **0.9744** | 0.9011 | **0.9520** | 0.8866 | 0.9493 | 0.8821 | 0.9503 | 0.8848 | 0.9400 | 0.8736 | 0.9331 | 0.8629 | 0.9241 | 0.8541 | 0.9537 | 0.8887 | 0.9529 | 0.8868 | 0.9053 | 0.8320 |
| General GNN | 0.9653 | 0.9015 | 0.9519 | 0.8907 | **0.9521** | **0.8921** | **0.9540** | **0.8953** | 0.9356 | 0.8735 | **0.9356** | **0.8792** | **0.9325** | **0.8625** | **0.9568** | **0.8957** | 0.9257 | 0.8526 | 0.9117 | 0.8426 |
| GINE | 0.9558 | **0.9132** | 0.9518 | **0.8939** | 0.9463 | 0.8878 | 0.9491 | 0.8914 | **0.9389** | 0.8748 | 0.9324 | 0.8589 | 0.9125 | 0.8429 | 0.9517 | 0.8878 | **0.9538** | **0.8928** | **0.9132** | **0.8448** |
| EdgeGNN | 0.9604 | 0.9055 | 0.9487 | 0.8851 | 0.9488 | 0.8884 | 0.9504 | 0.8891 | 0.9352 | **0.8765** | 0.9309 | 0.8575 | 0.9104 | 0.8410 | 0.9545 | 0.8914 | 0.9535 | 0.8897 | 0.9036 | 0.8345 |
| GraphTransformer | 0.9625 | 0.8950 | 0.9487 | 0.8751 | 0.9441 | 0.8742 | 0.9431 | 0.8763 | 0.9241 | 0.8333 | 0.9123 | 0.8592 | 0.9078 | 0.8309 | 0.9465 | 0.8769 | 0.9479 | 0.8817 | 0.8985 | 0.8256 |

| Methods | Amazon-Apps | | | | | | | | | | Amazon-Movie | | | | | | | | | |
|---|---|---|---|---|---|---|---|---|---|---|---|---|---|---|---|---|---|---|---|---|
| | Entangled-GPT | | GPT-3.5-TURBO | | BERT-Large | | BERT | | None | | Entangled-GPT | | GPT-3.5-TURBO | | BERT-Large | | BERT | | None | |
| | AUC | F1 | AUC | F1 | AUC | F1 | AUC | F1 | AUC | F1 | AUC | F1 | AUC | F1 | AUC | F1 | AUC | F1 | AUC | F1 |
| MLP | 0.8950 | 0.7980 | 0.8642 | 0.7752 | 0.8639 | 0.7698 | 0.8634 | 0.7698 | 0.8655 | 0.7738 | 0.8509 | 0.7490 | 0.8227 | 0.7269 | 0.8349 | 0.7553 | 0.8349 | 0.7555 | 0.8205 | 0.7317 |
| GraphSAGE | 0.8911 | 0.8073 | 0.8662 | 0.7853 | **0.8813** | 0.7971 | 0.8783 | 0.8015 | 0.8634 | 0.7366 | 0.8725 | 0.7911 | 0.8500 | 0.7665 | 0.9067 | 0.8298 | 0.9178 | 0.8426 | 0.8507 | 0.7591 |
| General GNN | **0.8956** | 0.8340 | **0.8810** | 0.8178 | 0.8768 | 0.8131 | 0.8757 | 0.8090 | **0.8680** | **0.8129** | **0.8849** | 0.8134 | **0.8659** | **0.7928** | **0.9206** | **0.8485** | 0.8937 | **0.8483** | 0.8617 | **0.7918** |
| GINE | 0.8875 | 0.8179 | 0.8559 | 0.8099 | 0.8680 | 0.8092 | 0.8555 | 0.8123 | 0.8671 | 0.8065 | 0.8712 | **0.8154** | 0.8603 | 0.7911 | 0.9187 | 0.8454 | 0.9165 | 0.8456 | 0.8591 | 0.7879 |
| EdgeGNN | 0.8956 | **0.8403** | 0.8720 | **0.8180** | 0.8813 | **0.8153** | **0.8804** | **0.8184** | 0.8520 | 0.8043 | 0.8708 | 0.8035 | 0.8565 | 0.7842 | 0.9171 | 0.8436 | **0.9181** | 0.8468 | 0.8552 | 0.7837 |
| GraphTransformer | 0.8634 | 0.7820 | 0.8395 | 0.7647 | 0.8748 | 0.7926 | 0.8736 | 0.7846 | 0.8469 | 0.7329 | 0.8537 | 0.7698 | 0.8339 | 0.7453 | 0.9035 | 0.8196 | 0.9044 | 0.8185 | 0.8393 | 0.7550 |

| Methods | Citation | | | | | | | | | | Twitter | | | | | | | | | |
|---|---|---|---|---|---|---|---|---|---|---|---|---|---|---|---|---|---|---|---|---|
| | Entangled-GPT | | GPT-3.5-TURBO | | BERT-Large | | BERT | | None | | Entangled-GPT | | GPT-3.5-TURBO | | BERT-Large | | BERT | | None | |
| | AUC | F1 | AUC | F1 | AUC | F1 | AUC | F1 | AUC | F1 | AUC | F1 | AUC | F1 | AUC | F1 | AUC | F1 | AUC | F1 |
| MLP | 0.9251 | 0.8679 | 0.9170 | 0.8598 | 0.9173 | 0.8561 | 0.8935 | 0.8613 | 0.8857 | 0.8015 | 0.7085 | 0.5669 | 0.6991 | 0.5430 | 0.8115 | 0.7898 | 0.8136 | 0.7148 | 0.7007 | 0.5430 |
| GraphSAGE | 0.9494 | 0.8972 | 0.9369 | 0.8758 | **0.9780** | **0.9300** | 0.8925 | 0.8345 | | | 0.6486 | 0.6193 | | | 0.8609 | 0.8177 | 0.7964 | | 0.5668 | 0.5940 |
| General GNN | 0.9470 | 0.8840 | 0.9258 | 0.8739 | 0.9281 | 0.8637 | 0.9327 | 0.8757 | 0.8984 | 0.8397 | **0.8118** | **0.7247** | 0.7888 | 0.7094 | 0.8531 | 0.7756 | 0.8062 | 0.6552 | 0.7017 | **0.6163** |
| GINE | **0.9538** | **0.9085** | **0.9482** | **0.8939** | 0.9443 | 0.8825 | 0.9736 | 0.9272 | 0.8744 | 0.8145 | 0.6835 | 0.6345 | 0.6696 | 0.6135 | 0.8306 | 0.7719 | 0.8738 | 0.7880 | **0.7213** | 0.6161 |
| EdgeGNN | 0.7382 | 0.5545 | 0.7136 | 0.5393 | 0.7132 | 0.5352 | 0.7401 | 0.6526 | 0.6965 | 0.5449 | 0.6940 | 0.6214 | 0.6854 | 0.6123 | 0.8290 | 0.6614 | 0.7513 | 0.6745 | 0.6124 | 0.5664 |
| GraphTransformer | 0.9536 | 0.8963 | 0.9350 | 0.8697 | 0.9439 | 0.8713 | **0.9789** | **0.9320** | **0.9172** | **0.8441** | 0.7030 | 0.6824 | 0.6859 | 0.6764 | **0.8967** | **0.8223** | **0.8768** | **0.8165** | 0.5908 | 0.5423 |

Table 3: Link prediction results for LLM as Predictor methods. The best method on each dataset is shown in bold.

| Methods | Goodreads-Children | | Goodreads-Crime | | Amazon-Apps | | Amazon-Movie | | Citation | | Twitter | |
|---|---|---|---|---|---|---|---|---|---|---|---|---|
| | AUC | F1 | AUC | F1 | AUC | F1 | AUC | F1 | AUC | F1 | AUC | F1 |
| GPT-3.5-TURBO | 0.4770 | 0.1413 | 0.4507 | 0.1104 | 0.5000 | **0.5200** | 0.4843 | 0.1342 | **0.8860** | **0.3514** | **0.4800** | 0.3312 |
| GPT-4 | **0.8780** | **0.6090** | **0.8890** | **0.6040** | **0.6212** | 0.1413 | 0.5000 | 0.3000 | 0.4735 | 0.3184 | 0.4300 | **0.6144** |

through PLMs. $k$ represents the layers of GNNs, $\mathcal{N}$ denotes the set of neighbors, $u$ denotes the target node, $\omega$ means the learning parameters in GNNs.

The advantage of this method over existing approaches is its ability to effectively preserve the semantic relationships between nodes and edges, making it more suitable for capturing complex relationships.

**LLM as Predictor**. Leveraging the robust text extraction capabilities of LLMs, LLMs can be directly employed to process raw text as textual prompt inputs to address graph-level task questions. Specifically, we can adopt a text template for each dataset to include the corresponding nodes and edges text to answer a given question, e.g. node classification or link prediction. We can formally define as follows:

$$A = f\{\mathcal{G}, Q\} \tag{4}$$

where $f$ is a prompt providing graph information. $\mathcal{G}$ represents a TEG and $Q$ is a question.

## 5 Experiments

In this section, we first introduce the detailed experimental settings in Section 5.1. Then, we conduct comprehensive benchmarks and perform a comprehensive analysis for link prediction and node classification in Section 5.2 and Section 5.3 respectively.

### 5.1 Experimental Settings

**Baselines. (1)** For the PLM-based Paradigm, we use three various sizes of PLM to encode texts in nodes for generating initial embeddings for nodes. These three models, representing large, medium, and small scales, include GPT-3.5-TURBO as the large-scale model, Bert-Large [7] as the medium-scale model, and Bert-Base [7] as the small-scale model. **(2)** For Edge-aware GNN-based methods, we select 5 popular GNN models: GraphSAGE [10], GeneralConv [48], GINE [14], EdgeGNN [43], and GraphTransformer [49]. We utilize three distinct scales of the PLMs, which are identical to those employed in the PLM-based paradigm, to encode text in nodes and edges. Afterward, these text embeddings on nodes and edges serve as their initial characteristics. **(3)** In the Entangled GNN-based approaches, the experimental setting is similar to Edge-aware GNN-based methods, with the key difference being the utilization of GPT-3.5-TURBO as the PLM to encode raw text in nodes and edges in an entangled manner. **(4)** For LLM-based Predictor methods, we chose state-of-the-art models GPT-3.5-TURBO and GPT-4, accessed via API, balancing performance and cost considerations.

Table 4: Node Classification ACC, Micro-AUC, Micro-F1 and F1 among PLM-based, GNN-based methods. AUC* and F1* represent Micro-AUC and Micro-F1 respectively. The best method for each PLM embedding on each dataset is shown in bold.

| Method | Children | | | | | | | | | | Crime | | | | | | | | | |
|---|---|---|---|---|---|---|---|---|---|---|---|---|---|---|---|---|---|---|---|---|
| | Entangled-GPT | | GPT-3.5-TURBO | | BERT-Large | | BERT | | None | | Entangled-GPT | | GPT-3.5-TURBO | | BERT-Large | | BERT | | None | |
| | AUC* | F1* | AUC* | F1* | AUC* | F1* | AUC* | F1* | AUC* | F1* | AUC* | F1* | AUC* | F1* | AUC* | F1* | AUC* | F1* | AUC* | F1* |
| MLP | 0.8785 | 0.5904 | 0.8505 | 0.5663 | 0.8593 | 0.5810 | 0.8597 | 0.5749 | 0.8452 | 0.5811 | 0.9253 | 0.6842 | 0.9149 | 0.6615 | 0.9150 | 0.6619 | 0.9151 | 0.6602 | 0.9154 | 0.6624 |
| GraphSAGE | **0.9569** | **0.8041** | 0.9342 | **0.7871** | **0.9162** | 0.7497 | **0.9152** | 0.7440 | **0.8713** | 0.6227 | 0.9663 | 0.8325 | **0.9549** | 0.8189 | 0.9445 | 0.7832 | 0.9463 | 0.7848 | 0.9221 | 0.7048 |
| General GNN | 0.9534 | 0.7942 | **0.9352** | 0.7846 | 0.9161 | **0.7502** | **0.9152** | 0.7451 | 0.8681 | 0.6162 | **0.9732** | **0.8437** | 0.9546 | **0.8200** | 0.9446 | **0.7854** | 0.9456 | **0.7888** | **0.9225** | **0.7262** |
| GINE | 0.9529 | 0.7930 | 0.9324 | 0.7777 | 0.9154 | 0.7466 | 0.9137 | **0.7552** | 0.8523 | 0.6558 | 0.9636 | 0.8260 | 0.9504 | 0.8073 | 0.9410 | 0.7766 | 0.9429 | 0.7852 | 0.9155 | 0.7117 |
| EdgeGNN | 0.9542 | 0.7890 | 0.9338 | 0.7808 | 0.9128 | 0.7463 | 0.9121 | 0.7452 | 0.8583 | 0.6466 | 0.9581 | 0.8179 | 0.9490 | 0.8052 | 0.9400 | 0.7657 | 0.9405 | 0.7726 | 0.9187 | 0.6830 |
| GraphTransformer | 0.9525 | 0.7902 | 0.9340 | 0.7823 | 0.9137 | 0.7497 | 0.9150 | 0.7491 | 0.8517 | **0.6565** | 0.9613 | 0.8322 | 0.9505 | 0.8151 | **0.9452** | 0.7795 | **0.9464** | 0.7834 | 0.9220 | 0.6944 |

| Method | Amazon-Apps | | | | | | | | | | Amazon-Movie | | | | | | | | | |
|---|---|---|---|---|---|---|---|---|---|---|---|---|---|---|---|---|---|---|---|---|
| | Entangled-GPT | | GPT-3.5-TURBO | | BERT-Large | | BERT | | None | | Entangled-GPT | | GPT-3.5-TURBO | | BERT-Large | | BERT | | None | |
| | AUC* | F1* | AUC* | F1* | AUC* | F1* | AUC* | F1* | AUC* | F1* | AUC* | F1* | AUC* | F1* | AUC* | F1* | AUC* | F1* | AUC* | F1* |
| MLP | 0.7750 | 0.3429 | 0.7520 | 0.3204 | 0.8935 | 0.4169 | 0.8970 | 0.3107 | 0.7352 | 0.3067 | 0.9736 | 0.5475 | 0.9618 | 0.5279 | 0.9752 | 0.5331 | 0.9750 | 0.5173 | 0.9493 | 0.4625 |
| GraphSAGE | **0.9439** | **0.4114** | **0.9274** | **0.3899** | **0.9226** | 0.3794 | **0.9229** | 0.3929 | **0.9161** | 0.3348 | 0.9764 | 0.5325 | 0.9674 | 0.5165 | **0.9773** | 0.4919 | **0.9771** | **0.5185** | 0.9681 | 0.5096 |
| General GNN | 0.9138 | 0.3806 | 0.8947 | 0.3604 | 0.9171 | 0.3817 | 0.9223 | 0.3803 | 0.9151 | **0.3932** | **0.9969** | 0.5301 | **0.9775** | 0.5156 | 0.9768 | 0.4827 | 0.9768 | 0.5006 | **0.9757** | 0.5115 |
| GINE | 0.9356 | 0.3862 | 0.9170 | 0.3588 | 0.9170 | 0.2623 | 0.9185 | 0.3592 | 0.9028 | 0.3507 | 0.9732 | 0.4531 | 0.9507 | 0.4246 | 0.9758 | 0.4781 | 0.9759 | 0.5085 | 0.9168 | 0.4127 |
| EdgeGNN | 0.8857 | 0.3749 | 0.8764 | 0.3477 | 0.8639 | 0.2739 | 0.8800 | 0.3063 | 0.8568 | 0.2247 | 0.9483 | 0.5224 | 0.9360 | 0.5060 | 0.9372 | 0.4672 | 0.9263 | 0.4743 | 0.9492 | 0.4853 |
| GraphTransformer | 0.9400 | 0.3772 | 0.9195 | 0.3548 | 0.9217 | 0.3425 | 0.9225 | 0.3818 | 0.9155 | 0.3860 | 0.9910 | 0.5285 | 0.9763 | 0.5175 | 0.9764 | 0.4856 | **0.9771** | 0.5124 | 0.9756 | **0.5126** |

| Method | Citation | | | | | | | | | | Twitter | | | | | | | | | |
|---|---|---|---|---|---|---|---|---|---|---|---|---|---|---|---|---|---|---|---|---|
| | Entangled-GPT | | GPT-3.5-TURBO | | BERT-Large | | BERT | | None | | Entangled-GPT | | GPT-3.5-TURBO | | BERT-Large | | BERT | | None | |
| | ACC | F1 | ACC | F1 | ACC | F1 | ACC | F1 | ACC | F1 | ACC | F1 | ACC | F1 | ACC | F1 | ACC | F1 | ACC | F1 |
| MLP | 0.7892 | 0.7879 | 0.7868 | 0.7859 | 0.7515 | 0.7471 | 0.8044 | 0.8032 | 0.7493 | **0.7471** | 0.8253 | 0.7549 | 0.8115 | 0.7261 | 0.8361 | 0.8193 | 0.8533 | 0.8329 | 0.8196 | 0.7383 |
| GraphSAGE | 0.7984 | **0.8144** | 0.7883 | 0.7874 | 0.7559 | 0.7525 | 0.8046 | 0.8060 | 0.7341 | 0.7308 | 0.8614 | 0.8055 | 0.8411 | 0.7903 | 0.8446 | 0.8305 | 0.8384 | 0.8247 | 0.8286 | 0.7802 |
| General GNN | **0.8079** | 0.8042 | 0.7906 | 0.7889 | 0.7546 | 0.7526 | 0.8057 | 0.8042 | 0.7361 | 0.7337 | **0.8725** | **0.8574** | **0.8610** | 0.8397 | 0.8368 | 0.8131 | 0.8609 | 0.8513 | 0.8401 | 0.8089 |
| GINE | 0.8055 | 0.8141 | **0.7934** | **0.7925** | 0.7599 | 0.7574 | 0.8106 | 0.8100 | 0.7316 | 0.7284 | 0.8649 | 0.8386 | 0.8438 | 0.8186 | 0.8401 | 0.8255 | 0.8460 | 0.8328 | 0.8254 | 0.7907 |
| EdgeGNN | 0.4261 | 0.3957 | 0.4140 | 0.3845 | 0.4082 | 0.3763 | 0.4200 | 0.3906 | 0.3935 | 0.3541 | 0.8714 | 0.8530 | 0.8551 | **0.8442** | 0.8649 | **0.8574** | **0.8694** | **0.8607** | **0.8529** | **0.8431** |
| GraphTransformer | 0.8022 | 0.7944 | 0.7903 | 0.7885 | 0.7531 | 0.7517 | 0.8070 | 0.8056 | 0.7369 | 0.7351 | 0.8720 | 0.8369 | 0.8563 | 0.8273 | 0.8342 | 0.8211 | 0.8402 | 0.8261 | 0.8197 | 0.7888 |

Table 5: Node Classification ACC, Micro-AUC, Micro-F1 and F1 for LLM as Predictor methods. AUC* and F1* represent Micro-AUC and Micro-F1 respectively. The best method on each dataset is shown in bold.

| Methods | Goodreads-Children | | Goodreads-Crime | | Amazon-Apps | | Amazon-Movie | | Citation | |
|---|---|---|---|---|---|---|---|---|---|---|
| | AUC* | F1* | AUC* | F1* | AUC* | F1* | AUC* | F1* | ACC | F1 |
| GPT-3.5-TURBO | 0.5200 | 0.0300 | 0.5400 | 0.0700 | **0.5000** | **0.0100** | 0.5159 | 0.0017 | 0.7098 | 0.3402 |
| GPT-4 | **0.6700** | **0.1800** | **0.6100** | **0.1400** | 0.4995 | 0.0002 | **0.5175** | **0.0029** | **0.8432** | **0.8450** |

**Implementation details.** We conduct experiments on 3 PLM-based, 18 GNN-based, and 2 LLM-based methods. For PLM-based methods, the dimensions of node embedding are 3072, 1024, and 768 generated by GPT-3.5-TURBO, Bert-Large, and Bert respectively. We set the MLP hidden layer to 2, with the number of hidden units in each layer being one-fourth of the units in the previous layer. For GNN-based methods, we adhere to the settings outlined in the respective paper. The parameters shared by all GNN models include dimensions of node and edge embeddings, model layers, and hidden units, with respective values set to 3072, 1024, and 768, as generated by GPT-3.5-TURBO, Bert-Large, and Bert, and 2, 256, respectively. We utilize cross-entropy loss with the Adam optimizer to train and optimize all the above models. The batch size is 1024. Each experiment is repeated three times. See Appendix B.1 for more details.

**Evaluations metrics.** We investigate the performance of different baselines through two tasks: link prediction and node classification. For the link prediction task, we use the Area Under ROC Curve (AUC) metric and F1 score to evaluate the model performance. For node classification, the choice of evaluation metrics depends on the nature of the classification tasks involved. In the context of datasets encompassing Goodreads-Children, Goodreads-Crime, and comics from Goodreads, along with Amazon-Apps and Amazon-Movie datasets from Amazon, the classification tasks involve multi-label node classification. Hence, metrics such as AUC-micro and F1-micro are chosen for evaluation. Conversely, datasets about citation networks and social networks are characterized by multi-class node classification, thus metrics such as ACC and F1 are selected for assessment.

## 5.2 Effectiveness Analysis for Link Prediction

In this subsection, we analyze the link prediction from the various models applied in the study. Table 2 and 3 represent the effect of link prediction on different datasets from various distinct models. The results on other datasets can be found in Appendix B.2. We can further draw several observations from Table 2 and 3. First, For PLM-based and GNN-based methods, the state-of-the-art methods for Goodreads-Children and Goodreads-Crime datasets are both GeneralConv. Under the condition of using the same embeddings, they outperform the worst method by approximately 5% and 7% in terms of AUC and F1 across these two datasets. For the Amazon-Apps and Amazon-Movie datasets, the state-of-the-art methods are EdgeGNN and GeneralConv. They outperform the worst method by approximately 3% and 7% in terms of AUC and F1 for Amazon-Apps, and by 8% and 7% in

terms of AUC and F1 for Amazon-Movie, respectively. For the Citation and Twitter datasets, the state-of-the-art method is GraphTransformer. It outperforms the worst method by approximately 20% and 30% in terms of AUC and F1 for Citation, and by 12% and 9% in terms of AUC and F1 for Twitter, respectively. Second, Entangled-GPT methods, which entangle edge text and node text first before encoding with GPT consistently outperform the approach of directly encoding the text through GPT, yielding about 2% improvement in both AUC and F1 metrics across all datasets on the link prediction tasks. Third, For the LLM as Predictor methods, we find that they do not perform well in predicting links. The best method among them has an AUC and F1 gap of approximately 10% - 30% compared to the best PLM-based and GNN-based methods for all datasets. Fourth, using edge text provides at least approximately a 3% improvement in AUC and at least approximately an 8% improvement in F1 compared to not using edge text for all datasets.

## 5.3 Effectiveness Analysis for Node Classification

In this subsection, we analyze the node classification results from various models. Table 4 and 5 display the impact on different datasets from various distinct, with additional results in Appendix B.3. We can derive some insights from the data. First, for PLM-based and GNN-based methods, the state-of-the-art models for Goodreads-Children and Goodreads-Crime are GraphSAGE and GeneralConv, respectively, outperforming the worst method by approximately 8% and 20% in AUC-micro and F1-micro for Goodreads-Children, and by 4% and 15% for Goodreads-Crime. In the E-commerce domain, GraphSAGE is the top method for Amazon-Apps and Amazon-Movie, outperforming the worst method by about 10% and 6% in AUC-micro and F1-micro for Amazon-Apps, and by 1% and 10% for Amazon-Movie. GINE and EdgeConv also show superior performance, exceeding the worst method by approximately 35% and 40% in ACC and F1 for Citation, and by 5% and 12% for Twitter. Second, Entangled-GPT methods outperform the approach of directly encoding the text through GPT, yielding about 2% improvement in both AUC and F1 metrics across all datasets on the node classification tasks. Third, LLM as Predictor methods perform poorly in node classification, with the best method showing an AUC-micro gap of about 30% compared to the best PLM-based and GNN-based methods. Their low F1-micro score could be due to the large number of predicted categories. Third, incorporating edge text results in at least a 3% improvement in AUC-micro and a 6% improvement in F1-micro across all datasets, compared to not using edge text.

**Observation.** *(1) The state-of-the-art model varies across different datasets.* Data variability and complexity play significant roles in influencing model performance. *(2) Edge text is crucial for TEG tasks.* Including edge text enriches relationship information, enabling a more precise depiction of interactions and relationships between nodes, which enhances overall model performance. *(3) Encoding text in an entangled manner is more beneficial for avoiding information loss.* The advantage of this method over existing approaches is its ability to effectively preserve the semantic relationships between nodes and edges, making it more suitable for capturing complex relationships. *(4) The scale of PLMs significantly impacts the performance of TEG tasks, especially on datasets with rich text on nodes and edges.* Larger model scales result in higher-quality text embeddings and better semantic understanding, leading to improved model performance. *(5) When using LLMs as predictors, they struggle to fully comprehend graph topology information.* LLMs are designed for linear sequence data and do not inherently capture the complex relationships and structures present in graph data, leading to lower performance on TEGs link prediction and node classification.

## 5.4 Parameter Sensitivity Analysis

We further analyze the impact of text embeddings generated from PLMs. For the link prediction task, as shown in Table 2, using small-scale PLMs like BERT improves the AUC and F1 scores by approximately 5% compared to not using text embeddings. Medium-scale models such as BERT-Large and large-scale models like GPT-3.5-TURBO improve the AUC and F1 scores by about 7% across all datasets. For node classification, as shown in Table 4, the improvement is slightly less pronounced. Small-scale PLMs like BERT improve the AUC-micro and F1-micro scores by approximately 3%, while medium-scale models like BERT-Large and large-scale models like GPT-3.5-TURBO improve these scores by about 3.5% across all datasets.

# 6 Discussion

Textual-Edge graphs have emerged as a prominent graph format, which finds extensive applications in modeling real-world tasks. Our research focuses on comprehensively understanding the textual

attributes of nodes and their topological connections. Furthermore, we believe that exploring strategies to enhance the efficiency of LLMs in processing TEGs is deemed meaningful. Despite the proven effectiveness of LLMs, their operational efficiency, especially in managing TEGs, poses a significant challenge. Notably, employing APIs like GPT4 for extensive graph tasks may result in considerable expenses under current billing models. Additionally, deploying open-source large models such as LLaMa for tasks like parameter updates or inference in local environments demands substantial computational resources and storage capacity. Please refer to the Appendix C for more details.

## 7 Conclusion

We introduce the inaugural TEG benchmark, TEG-DB, tailored to delve into graph representation learning on TEGs. It incorporates textual content on both nodes and edges compared to traditional TAG with only node information. We gather and furnish nine comprehensive textual-edge datasets to foster collaboration between the NLP and GNN communities in exploring the data collectively. Our benchmark offers a thorough assessment of various learning approaches, affirming their efficacy and constraints. Additionally, we plan to persist in uncovering and building more research-oriented TEGs to further propel the ongoing robust growth of the domain.

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

# A   Datasets

## A.1   Dataset format

For each dataset, all unprocessed raw files are represented in .json format. After preprocessing, we store the graph-type data compatible with PyTorch Geometric (PyG) [8] in the .pt format using PyTorch. Specifically, we have retained the raw text on nodes, the labels on nodes, the raw text on edges, and the adjacency matrix. We uniformly store the text embeddings of node and edge text in .npy files and load them during data processing.

## A.2   Datasets license

The datasets are subject to the MIT license. For precise license information, please refer to the corresponding GitHub repository.

# B   Experiment

## B.1   Implementation Details

GNNs are mainly derived from the implementation in the PyG library [8]. For the node classification task, numerical node labels corresponding to the nodes within the graph are necessary. This involves converting the categorical node categories found in the original data into numerical node labels within the graph. For the link prediction, we randomly sample node pairs that do not exist in the graph as negative samples, along with some edges present as positive samples. For LLM-based predictor methods, we focus on node classification and link prediction tasks. For node classification, inspired by the recent LLM-based classification algorithm [27], we use GPT-4 and GPT-3.5-TURBO models to predict the classification of text nodes by providing the probability for each class. We randomly select 1,000 text nodes along with all classification labels for this task. For the link prediction task, we also apply the GPT-4 and GPT-3.5-TURBO models to determine whether two text edges are related, providing an answer with the corresponding probability. For this task, we randomly select 1,000 pairs of positive text edge indices from the graph and an equal number of negative edges.

## B.2   Effectiveness Analysis for Link Prediction

In this subsection, we further analyze the link prediction from the various models applied in the study. Table 6 and 7 represent the effect of link prediction on different datasets from various distinct. We can further draw several observations from Table 6 and 7. First, For PLM-based and GNN-based methods, the state-of-the-art methods for Goodreads-Comics and Goodreads-History datasets are GeneralConv and GINE, respectively. Under the condition of using the same embeddings, they outperform the worst method by approximately 6% and 7% in terms of AUC and F1 across these two datasets. For the Reddit dataset, the state-of-the-art method is GeneralConv. It outperforms the worst method by approximately 3% and 5% in terms of AUC and F1, respectively. Second, for the LLM as a predictor method, we find that they do not perform well in predicting links. The best method among them has an AUC and F1 gap of approximately 10% - 30% compared to the best PLM-based and GNN-based methods for all datasets. Third, Using edge text provides at least approximately a 3% improvement in AUC and at least approximately an 8% improvement in F1 compared to not using edge text for all datasets.

## B.3   Effectiveness Analysis for Node Classification

In this subsection, we further analyze the node classification results from various models. Table 8 and 9 display the impact on different datasets. We can derive some insights. First, for PLM-based and GNN-based methods, the state-of-the-art models for Goodreads-Comics and Goodreads-History are GeneralConv and GINE, respectively, outperforming the worst method by approximately 8% and 15% in AUC-micro and F1-micro for Goodreads-Comics, and by 6% and 9% for Goodreads-History. GraphTransformer outperforms the worst method by approximately 2% and 1% in ACC and F1 for Citation. Second, LLM as Predictor methods perform poorly in node classification, with the best method showing an AUC-micro gap of about 20% compared to the best PLM-based and GNN-based methods. Their low F1-micro score could be due to the large number of predicted categories. Third, incorporating edge text results in at least a 3% improvement in AUC-micro and a 6% improvement in F1-micro across almost all datasets, compared to not using edge text.

Table 6: Link prediction AUC and F1 among PLM-based, GNN-based methods. The best method for each PLM embedding on each dataset is shown in bold.

| Methods | Goodreads-Comics | | | | | | | | Goodreads-History | | | | | |
|---|---|---|---|---|---|---|---|---|---|---|---|---|---|---|
| | GPT-3.5-TURBO | | BERT-Large | | BERT | | None | | BERT-Large | | BERT | | None | |
| | AUC | F1 | AUC | F1 | AUC | F1 | AUC | F1 | AUC | F1 | AUC | F1 | AUC | F1 |
| MLP | 0.8902 | 0.8136 | 0.8900 | 0.8130 | 0.8900 | 0.8128 | 0.8928 | 0.8167 | 0.8922 | 0.8897 | 0.8923 | 0.8897 | 0.8913 | 0.8149 |
| GraphSAGE | 0.9406 | 0.8689 | 0.9511 | 0.8854 | 0.9537 | 0.8860 | 0.9403 | 0.8732 | 0.9587 | 0.8702 | 0.9591 | 0.8698 | 0.9053 | 0.8320 |
| GeneralConv | 0.9478 | 0.8843 | **0.9535** | **0.8930** | **0.9544** | **0.8942** | **0.9458** | **0.8825** | 0.9624 | **0.8900** | 0.9629 | 0.8897 | 0.9117 | 0.8426 |
| GINE | **0.9489** | **0.8870** | 0.9480 | 0.8857 | 0.9471 | 0.8833 | 0.9446 | 0.8819 | **0.9631** | 0.8669 | **0.9634** | **0.8937** | **0.9132** | **0.8448** |
| EdgeConv | 0.9448 | 0.8819 | 0.9495 | 0.8867 | 0.9477 | 0.8853 | 0.9444 | 0.8810 | 0.9457 | 0.8695 | 0.9456 | 0.8650 | 0.9036 | 0.8345 |
| GraphTransformer | 0.9380 | 0.8687 | 0.9433 | 0.8747 | 0.9466 | 0.8781 | 0.9362 | 0.8661 | 0.9589 | 0.8698 | 0.9590 | 0.8690 | 0.8985 | 0.8256 |

| Methods | Reddit | | | | | | | |
|---|---|---|---|---|---|---|---|---|
| | GPT-3.5-TURBO | | BERT-Large | | BERT | | None | |
| | AUC | F1 | AUC | F1 | AUC | F1 | AUC | F1 |
| MLP | 0.9909 | 0.9651 | 0.9866 | 0.9576 | 0.8900 | 0.8128 | 0.8928 | 0.8167 |
| GraphSAGE | 0.9908 | 0.9810 | 0.9897 | 0.9800 | 0.9537 | 0.8860 | 0.9403 | 0.8732 |
| GeneralConv | **0.9964** | 0.9809 | 0.9956 | **0.9815** | 0.9544 | 0.8942 | 0.9458 | 0.8825 |
| GINE | 0.9962 | 0.9809 | **0.9958** | 0.9801 | 0.9471 | 0.8833 | 0.9446 | 0.8819 |
| EdgeConv | 0.9926 | **0.9818** | 0.9926 | 0.9803 | 0.9477 | 0.8853 | 0.9444 | 0.8810 |
| GraphTransformer | 0.9944 | 0.9810 | 0.9940 | 0.9803 | 0.9466 | 0.8781 | 0.9362 | 0.8661 |

Table 7: Link prediction results for LLM as Predictor methods. The best method on each dataset is shown in bold.

| Methods | Goodreads-Comics | | Goodreads-History | | Reddit | |
|---|---|---|---|---|---|---|
| | AUC | F1 | AUC | F1 | AUC | F1 |
| GPT-3.5-TURBO | 0.4565 | **0.3588** | 0.6031 | 0.5234 | 0.4980 | 0.3440 |
| GPT-4 | **0.5446** | 0.2461 | **0.8661** | **0.8685** | **0.6632** | **0.6478** |

Table 8: Node Classification ACC, Micro-AUC, Micro-F1 and F1 among PLM-based, GNN-based methods. AUC* and F1* represent Micro-AUC and Micro-F1 respectively. The best method for each PLM embedding on each dataset is shown in bold.

| Methods | Goodreads-Comics | | | | | | | | Goodreads-History | | | | | |
|---|---|---|---|---|---|---|---|---|---|---|---|---|---|---|
| | GPT-3.5-TURBO | | BERT-Large | | BERT | | None | | BERT-Large | | BERT | | None | |
| | AUC* | F1* | AUC* | F1* | AUC* | F1* | AUC* | F1* | AUC* | F1* | AUC* | F1* | AUC* | F1* |
| MLP | 0.8361 | 0.5117 | 0.8360 | 0.5211 | 0.8370 | 0.5214 | 0.8373 | 0.5214 | 0.7831 | 0.8099 | 0.7825 | 0.8097 | 0.7824 | 0.8096 |
| GraphSAGE | 0.9068 | 0.7379 | 0.8965 | 0.7118 | 0.8965 | 0.7088 | 0.8689 | 0.6401 | 0.8543 | 0.8975 | 0.8538 | 0.8970 | 0.8044 | 0.8088 |
| GeneralConv | **0.9107** | **0.7455** | 0.8982 | 0.7134 | **0.8991** | 0.7116 | **0.8739** | 0.6541 | 0.8543 | 0.8986 | 0.8538 | 0.8981 | 0.8119 | 0.8126 |
| GINE | 0.9006 | 0.7187 | 0.8943 | 0.7084 | 0.8932 | 0.7140 | 0.8627 | 0.6457 | 0.8541 | **0.9015** | 0.8549 | **0.9022** | **0.8133** | **0.8226** |
| EdgeConv | 0.9015 | 0.7127 | 0.8923 | 0.7066 | 0.8931 | 0.7089 | 0.8648 | 0.6260 | 0.8520 | 0.8974 | 0.8515 | 0.8960 | 0.8059 | 0.8116 |
| GraphTransformer | 0.9027 | 0.7285 | 0.8940 | **0.7175** | 0.8966 | **0.7151** | 0.8704 | **0.6554** | **0.8555** | 0.9009 | **0.8647** | 0.8995 | 0.8101 | 0.8089 |

| Methods | Reddit | | | | | | | |
|---|---|---|---|---|---|---|---|---|
| | GPT-3.5-TURBO | | BERT-Large | | BERT | | None | |
| | ACC | F1 | ACC | F1 | ACC | F1 | ACC | F1 |
| MLP | 0.9839 | 0.9817 | 0.9793 | 0.9774 | 0.9803 | 0.9784 | 0.9795 | 0.9779 |
| GraphSAGE | 0.9974 | 0.9962 | **0.9975** | 0.9964 | 0.9974 | 0.9965 | **0.9974** | **0.9965** |
| GeneralConv | **0.9975** | **0.9966** | 0.9974 | 0.9963 | 0.9973 | 0.9964 | 0.9973 | 0.9964 |
| GINE | 0.9973 | 0.9962 | 0.9973 | 0.9963 | **0.9974** | 0.9965 | **0.9974** | 0.9962 |
| EdgeConv | 0.9973 | 0.9960 | 0.9973 | 0.9960 | 0.9973 | 0.9960 | 0.9973 | 0.9959 |
| GraphTransformer | 0.9973 | 0.9963 | 0.9974 | **0.9965** | 0.9974 | **0.9966** | 0.9973 | 0.9964 |

Table 9: Node Classification ACC, Micro-AUC, Micro-F1 and F1 for LLM as Predictor methods. AUC* and F1* represent Micro-AUC and Micro-F1 respectively. The best method on each dataset is shown in bold.

| Methods | Goodreads-Comics | | Goodreads-History | | Reddit | |
|---|---|---|---|---|---|---|
| | AUC* | F1* | AUC* | F1* | ACC | F1 |
| GPT-3.5-TURBO | 0.4900 | 0.0400 | 0.6827 | 0.4147 | 0.8625 | 0.9262 |
| GPT-4 | **0.5600** | **0.0600** | **0.8202** | **0.7394** | **0.9767** | **0.9882** |

## C  Discussion

Notably, employing APIs like GPT4 for extensive graph tasks may result in considerable expenses under current billing models. Additionally, deploying open-source large models such as LLaMa for tasks like parameter updates or inference in local environments demands substantial computational resources and storage capacity. Consequently, enhancing the efficiency of LLMs for graph-related tasks remains a critical concern. Moreover, the constraints imposed by context windows in LLMs also impact their effectiveness in encoding node and edge text within TEGs.

## D  Limitation

Comprehensive evaluation of tasks often demands significant computational resources, which can be a burden for researchers and smaller organizations.

