# OpenReview forum: "TEG-DB: A Comprehensive Dataset and Benchmark of Textual-Edge Graphs"
_NeurIPS.cc/2024/Datasets_and_Benchmarks_Track — NeurIPS 2024 Track Datasets and Benchmarks Poster_

### Official Review · Reviewer_crpc · 2024-07-20
**Review for TEG-DB**

**Rating:** 7
**Confidence:** 3
**Correctness:** The claims should be correct.
**Clarity:** The paper is clearly written.

**Review:**

- Quality

The paper is of decent quality with the inclusion of datasets from various domains: citation networks, social networks, and e-commerce graphs, which ensures broad applicability of the benchmark.
A standardized pipeline of data processing, loading, and evaluation is provided. Models including GNNs and PLMs are benchmarked.

- Clarity

The paper is clearly written. The benefits of incorporating textual information on edges are demonstrated.

- Originality

To my knowledge, introducing graph datasets with rich textual descriptions on both nodes and edges is a novel contribution, filling a gap in the existing TAG datasets.

- Significance

The introduction of TEG-DB has the potential to advance the field of graph representation learning by providing a comprehensive benchmark for future research.

By offering an open-source resource, the paper encourages collaboration within the research community.

**Strengths:**

The paper's strengths lie in its originality and quality as discussed in the review above. Especially originality, to my knowledge, this is the first open graph benchmark with rich textural edge features.

**Additional Feedback:**

No.

**Documentation:**

Yes. A Github repo is provided.

**Ethics:**

No potential ethics issues.

**Limitations:**

The paper can be further improved by supporting other GNN platforms and benchmarking with more advanced GNN models.

**Opportunities For Improvement:**

- I checked the Github repo, and it seems like PyG is the only GNN platform supported. Will other platforms like DGL be supported in the future?

- In the benchmarking part, the GNN models considered are not the latest ones, like GraphSAGE. Including more recent GNNs could improve future versions.

**Relation To Prior Work:**

Yes. The differences from previous contributions (TAGs) are discussed.

**Summary And Contributions:**

This paper addresses the limitations of existing Text-Attributed Graphs by introducing a new dataset and benchmark collection called TEG-DB. Traditional TAG datasets often feature textual information only at the nodes. Edges either have no features or only categorical attributes. This paper emphasizes the importance of rich textual annotations on both nodes and edges for exploring contextual relationships between entities. TEG-DB offers a range of datasets across domains, including citation, social, and Amazon graphs.

TEG-DB introduces a standardized pipeline for data preprocessing, loading, and evaluation. The paper aims to advance the textual-edge graph research by leveraging methodologies that exploit rich textual node and edge descriptions to enhance graph analysis. The project is made available on Github.

---

> ### Author Rebuttal · Authors · 2024-08-17
>
> ### Q1. Other platforms like DGL will be supported in the future
> Yes, we are expanding our benchmarks to other platforms such as DGL.
>
> ### Q2. In the benchmarking part, the GNN models considered are not the latest ones, like GraphSAGE. Including more recent GNNs could improve future versions.
>
> We have already included GraphSAGE model in our experiment as shown in Table 2 and Table 4 in the paper. Moreover, to further address your suggestion, **we conducted additional experiments by adding mainstream GNNs using Bert text embedding, including GCN, GAT, RevGAT. The F1 score results of the five datasets are shown in the table.** We find that (1) For link prediction, GCN generally performs better than GAT and RevGAT across most datasets, except for the Twitter dataset, where RevGAT shows a notable improvement. (2) For node classification,  GCN, GAT, and RevGAT achieve similar performance for node classification on the Twitter, and Amazon-Movie datasets, but their performance varies more significantly on the Reddit, Amazon-Apps and Children datasets.
>
> | Link Prediction | Reddit | Amazon-Apps  | Twitter | Amazon-Movie  | Children  |
> |-----------------|--------|--------------|---------|--------|-----------|
> | GCN             | 0.9720 | 0.7798       | 0.5000  | 0.7868 | 0.8619    |
> | GAT             | 0.9546 | 0.7455       | 0.5000  | 0.7881 | 0.8398    |
> | RevGAT          | 0.9801 | 0.7422       | 0.5945  | 0.7882 | 0.8398    |
>
>
>
> | Node Classification | Reddit | Amazon-Apps  | Twitter | Amazon-Movie  | Children  |
> |---------------------|--------|--------------|---------|--------|-----------|
> | GCN                 | 0.9532 | 0.1625       | 0.8144  | 0.9962 | 0.7024    |
> | GAT                 | 0.9548 | 0.2047       | 0.8141  | 0.9962 | 0.7263    |
> | RevGAT              | 0.9723 | 0.1514       | 0.8146  | 0.9962 | 0.7263    |

---

### Official Review · Reviewer_bd5x · 2024-07-22
**Initial review**

**Rating:** 6
**Confidence:** 5
**Clarity:** See more details in my Review section.

**Review:**

**Quality and Clarity:**

Overall, the quality and clarity of this manuscript is good. However, a deeper analysis of the methods and results used can be made to improve the quality and clarity.

**Originality:**

This work is innovative since it noticed the big gap of TAGs that lack the usage of textual information of edges and built the foundation for TEGs.

**Significance:**

By pointing out the limitations of TAGs and making the benchmark for TEGs, this work lays the groundwork for future exploration of TEGs as well as more applications of graphs in real-world settings.

**Pros:**
The paper is well structured and formatted. It effectively introduces the background, identifies limitations in existing works, articulates the problem statement, and conveys the motivation clearly, which ensures a thorough understanding of the research objectives and their potential impact.
The work addresses a significant limitation in current TAG datasets where edges lack the corporation of textual attributes, thus hindering the exploration of more complex relationships between entities. The introduction of textual edge annotations is a novel and promising approach to overcome this limitation.

**Cons:**
Insufficient Explanation of Results

In Section 5.2 and Section 5.3, the observations and results are presented without sufficient explanation of why certain methods perform better or worse. Providing insights or hypotheses about the performance differences would add depth to the analysis.

Overuse of Numerical Data and Repeated Phrases
The subsections of Experiments list too many specific numerical improvements (e.g., percentages for AUC and F1), which can be overwhelming. Summarizing key improvements and providing detailed numbers in a table or restructuring the results information would enhance readability.

Lack of Adapted Solutions
It is confusing that the title of Section 4.3 is “Adapting Existing Methods to Solve Problems in TEGs”, but this study only mentions limitations without proposing concrete adapted solutions.

**Strengths:**

By offering comprehensive TEG datasets across diverse domains and conducting extensive benchmark experiments, this work significantly contributes to advancing research in textual-edge graph analysis. This relevance is underscored by its potential to impact various applications, from citation networks to social networks, thereby appealing to a broad spectrum of researchers interested in graph-based methodologies.

The detailed analysis of TEG-based methods, including the evaluation of different models and embedding, guides future research directions. This aspect is crucial for researchers aiming to leverage rich textual information for enhanced graph analysis in complex real-world networks.

**Additional Feedback:**

See review for more details.

**Correctness:**

The details of constructed datasets are correct and most of the evaluation methods and experimental design are appropriate, except for the following two questions:

- Can you provide some qualitative analysis to demonstrate the significance of textual edges in some datasets? For example, in social networks, it would be better to provide an example to show the difference between using binary edges and textual edges.
-

**Documentation:**

The paper includes sufficient details in terms of data collection and organization.

**Limitations:**

See the review and opportunities for improvement for more details.

**Opportunities For Improvement:**

**Insufficient Explanation of Results: **
- For example, It says “for PLM-based and GNN-based methods, the state-of-the-art methods for Goodreads-Children and Goodreads-Crime datasets are GeneralConv and GINE, respectively.” How it is concluded that the state-of-the-art method for Goodreads-Crime is GINE since GeneralConv achieves the best performance with the embedding of GPT-3.5-TURBO and BERT-Large on the Goodreads-Crime dataset.

- For example, the result “Under the condition of using the same embeddings, they outperform the worst method by approximately 5% and 7% in terms of AUC and F1 across these two datasets.” is ambiguous. What is the worst method on the Goodreads-Crime dataset and on Goodreads-Children? Since there are not only two metrics but also two datasets, what specifically do 5% and 7% refer to? More details should be added to explain the results.

- The authors list the comparison results among all these methods. How about a deeper understanding or explanation of why such a method outperformed others? e.g., why does GINE achieve the best performance on the Citation dataset?

**Overuse of Numerical Data and Repeated Phrases**

- For example, the text contains redundant phrases (e.g., "approximately number% and number%") repeated multiple times, making it hard to understand the context. One possible solution is using figures to improve conciseness and readability.

**Lack of Adapted Solutions**

- The title of section 4.3 is named “Adapting Existing Methods to Solve Problems in TEGs”, but what are the problems of TEGs being solved here? And how the existing methods are adapted? The authors only mentioned the limitations or challenges of PLM-based and GNN-based methods in processing textual-edge graphs but didn’t mention how these methods are adapted or refined to solve problems in TEGs.

**Writing Issues:**
- Page 8, Section 5.2: “First, For PLM-based ...” should be “First, for PLM-based ...”
- Page 8, Section 5.2: “Third, Using edge ...” should be “Third, using edge ...”
- Table 3 is not referenced in the context.
- Table 4, Citation dataset: the best results should be highlighted in bold in the “w/o edge text” column.

Refined Definition for LLM as Predictor: The current definition of LLM as Predictor lacks clarity. What does “A” refer to? How LLM is directly utilized to process raw text for graph-level tasks in the given definition. Clarifying this aspect is crucial to better understand how LLM works as a predictor.

Answering the following questions can better improve the manuscript:
- For the PLM-based paradigm, how textual edges are applied based on Eq.(1)?
- Can you provide some qualitative analysis to demonstrate the significance of textual edges in some datasets? For example, in social networks, it would be better to provide an example to show the difference between using binary edges and textual edges.
- Can incorporating textual-edge information further improve other graph-based tasks except for link prediction and node classification? Would you mind providing your understanding and making an analysis based on the observation from current work?

**Relation To Prior Work:**

The paper clearly discussed the difference from previous paper, which is the incorporation of edge features in the textual-attributed graphs.

**Summary And Contributions:**

Summary: Overall, this research introduced TEG datasets and conducted extensive experiments to advance TEG research by developing methods that leverage detailed textual descriptions for enhanced graph analysis and insights.

Contributions:

(1) This research introduced the first open dataset and benchmark specifically designed for TEGs.

(2) A standardized pipeline for TEG research was proposed, encompassing data preprocessing, data loading, and model evaluation.

(3) Extensive benchmark experiments were conducted, providing insights through analysis of results obtained from various TEG-based methods.

---

> ### Author Rebuttal · Authors · 2024-08-17
>
> ### Q1. Insufficient Explanation of Results:
> We identified the state-of-the-art model for each dataset by averaging the performance of each GNN model across different LLM embeddings.
>
> The "worst method" is the GNN model with the lowest average performance across LLM embeddings for the same dataset. The state-of-the-art models outperform the worst method by roughly 5% and 7% in AUC and F1 scores on the Goodreads-Crime and Goodreads-Children datasets, respectively.
>
> **Insights**: 1. GNN models with more parameters, like GINE and GeneralConv [1], tend to perform better because the complex semantic relationships in node and edge text require substantial capacity to be effectively captured. The additional parameters enable more precise and expressive feature representations, enhancing the model's ability to handle the incremental information from textual edges and their complex interplay with node text.  2. Models that better treat node and edge text as a unified whole during the message-passing phase tend to achieve better performance. For example, GINE outperforms EdgeConv in this regard.
>
> [1] You J, Ying Z, Leskovec J. Design space for graph neural networks[J]. Advances in Neural Information Processing Systems, 2020, 33: 17009-17021.
>
> ### Q2 Overuse of Numerical Data and Repeated Phrases
> Thank you for pointing out. We have reflected the changes in writing for a clear purpose.
>
> ### Q3. Lack of Adapted Solutions
> Problems: As mentioned in the introduction, TEGs are characterized by rich texts on edges and nodes, as well as complex semantic relationships between nodes, edges, and their interplays. The main challenges addressed here are how to fully understand and capture these semantic relationships while also considering the topological structure of the TEG graph.
>
> **How adapted:** 1. To obtain the initial features for nodes and edges, we use PLM-based embeddings for the text on nodes and edges, as traditional methods often rely on shallow encoders like Skip-Gram and GloVe.  PLMs can understand and encode complex semantic relationships, improving the quality of the node and edge features. 2.  Some GNN methods, such as GraphSAGE, do not directly handle edge features. To address this, we linearly map the edge attributes and incorporate them into the node representation during message passing. This adaptation allows us to better capture and utilize the textual information and structural relationships in TEGs.
>
> ### Q4.  How textual edges are applied in PLM methods
> We directly map the edge embedding using a linear transformation and then add them to the node representation, which is input into the MLP.
>
> ### Q5.  Can you provide some qualitative analysis to demonstrate the significance of textual edges in some datasets?
> For a qualitative analysis demonstrating the significance of textual edges, please refer to Table 2 and Table 4 in the paper. The last column of each dataset in these tables shows the impact of excluding edge text. These results highlight the significant role that edge text plays.  **For using binary edges and textual edges in social network, below are the F1 score results on social network with  GPT text embedding.**
> We can clearly observe that edge text plays a crucial role in enhancing the understanding of the relationships between nodes within a network. By providing context and additional semantic information about the connections, edge text helps to capture the nuances of these relationships, which can significantly improve the performance of models in tasks such as link prediction and node classification.
>
> | Social Network link prediction | Textual Edge | Binary Edge |
> |--------------------------------|--------------|-------------|
> | GraphSAGE                      | 0.6193       | 0.5952      |
> | GeneralConv                    | 0.7094       | 0.6179      |
> | GINE                           | 0.6135       | 0.6012      |
> | EdgeConv                       | 0.6123       | 0.5769      |
> | GraphTransformer               | 0.6764       | 0.6023      |
>
> | Social Network node classification | Textual Edge | Binary Edge |
> |---|---|---|
> | GraphSAGE | 0.7903 | 0.7826 |
> | GeneralConv | 0.8397 | 0.8172 |
> | GINE | 0.8186 | 0.8107 |
> | EdgeConv | 0.8442 | 0.8436 |
> | GraphTransformer | 0.8273 | 0.7982 |
>
> ### Q5. Textual-edge information function
>
> Incorporating textual-edge information has the potential to enhance a variety of graph-based tasks beyond link prediction and node classification. For example, in tasks like graph clustering, textual-edge information can provide additional context for grouping nodes with similar textual relationships, leading to more semantically meaningful clusters. Similarly, in graph summarization, textual-edge data can contribute to more informative summaries by highlighting key connections and relationships that are otherwise missed in binary or feature-only edge representations.
> Our observations indicate that incorporating textual edges has significantly improved node classification and link prediction tasks. This suggests that the rich semantic information in textual edges can enhance other graph-based tasks by providing a deeper understanding of relationships within the graph. We believe further exploration of textual-edge integration in these tasks could yield promising results.

---

> > ### Comment · Reviewer_bd5x · 2024-08-28
> > **Thank you for the reply**
> >
> > Thank you for the reply. The results comprehensively indicate the importance of considering the text information in the graph machine learning tasks.
> >
> > One specific follow-up question comparing the attached results between link prediction and node classification indicates that incorporating textual features leads to a significant performance boost on link prediction rather than node classification. Is there a potential reason for that?

---

> > ### Author Rebuttal · Authors · 2024-08-29
> >
> > ### A potential reason for a significant performance boost on link prediction rather than node classification
> > In link prediction tasks, since the goal is to predict the existence of a link between two nodes, representations of two connecting nodes are utilized. This contrasts with node classification tasks, where only the representation of a single node is considered. As a result, link prediction tends to leverage more edge textual information because it accounts for the interaction between nodes. Therefore, incorporating edge textual features could potentially lead to a more significant performance improvement in link prediction tasks.

---

> > > ### Comment · Reviewer_bd5x · 2024-08-31
> > > **Follow-up**
> > >
> > > Thank you for the discussion. That could be reasonable.

---

### Official Review · Reviewer_d2Yp · 2024-07-25

**Rating:** 6
**Confidence:** 4
**Correctness:** Yes
**Clarity:** Yes

**Review:**

This work is well organized and innovates in the new datasets and corresponding benchmark experiments. Specifically, the pros and cons are listed below.

Pros.
1. TEG-DB is the first open textual-edge graph datasets, including 9 datasets from 4 diverse domains.
2.	To benchmark existing methods, the authors develop a standardized pipeline from the data preprocessing, data loading, and model evaluation aspects.
3.	Extensive benchmark experiments are performed to assess the efficacy of different models, embedding strategies.

Cons.
1. While the authors describe the data preparation and construction steps in Section 4.2, it seems all the collected datasets are just borrowed from previous datasets. What’s the main challenges or endeavors made by this work to collect TEG datasets?
2. The difference between TEG and existing heterogeneous datasets is not well explained. For example, what’s the key difference between Citation graph and MAG or ogbn-paper datasets?

**Strengths:**

1. The introduced TEG datastes may be interesting to a wide range of researchers in graph machine learning.
2. The standardized pipeline is crucial for comprehensive and fair comparison in the literature.

**Additional Feedback:**

Please see the comments above.

**Documentation:**

Yes

**Limitations:**

Yes

**Opportunities For Improvement:**

Please see the comments above.

**Relation To Prior Work:**

Good but not in a comprehensive way.

**Summary And Contributions:**

This paper focuses on text-attributed graphs and introduces textual-edge graphs datasets and benchmarks (TEG-DB), a comprehensive and diverse collection of benchmark textual-edge datasets. Additionally, they conduct extensive benchmark experiments on TEG-DB to evaluate the current progress.

---

> ### Author Rebuttal · Authors · 2024-08-17
>
> ### Q1. Main challenges or endeavors made by this work to collect TEG datasets?
> 1. We have introduced new datasets that were completely collected and established by us. For example, the citation network dataset was crawled from the Semantic Scholar website with the code we developed. Specifically, we crawled the website comprehensively including title, abstract, paper body, citation content and links, and author information. Among them, paper titles, and abstracts are formulated as node text, while citation content and links are formulated as edge texts.
> 2. Previous datasets were merely sequential text without any graph structure and we established a unified code to thoroughly re-process the data into  Textual-Edge Graphs (TEG)
> 3. We standardized these datasets into a consistent format, aligned with our pipeline for preprocessing, data loading, and model evaluation, providing the research community with a more accessible and uniform resource for TEG studies.
> ### Q2. Difference between TEG and existing heterogeneous datasets?
> **The key difference between TEG and existing heterogeneous datasets lies in the nature of the textual information associated with edges.** In TEG, each edge is accompanied by rich textual data, which is not a feature in the mentioned heterogeneous datasets. On the other hand, TEG does not need to have heterogeneous nodes or edges. While Citation graphs, MAG, and ogbn-paper datasets may include feature attributes associated with nodes, they typically lack the rich textual content on edges that is central to TEG. This difference is crucial as it fundamentally alters how models must be designed and evaluated, with a greater emphasis on effectively leveraging and processing the textual information in TEG.

---

### Official Review · Reviewer_r2GB · 2024-07-27

**Rating:** 6
**Confidence:** 4

**Review:**

Quality: The quality of this paper is fair. The construction of the dataset part is good, but both the writing and experimental parts of the paper should be improved.
Clarity: The writing of this paper could be improved. More specific comments can be found in the latter section.
Originality: The motivation and methodology for the construction of the TEG dataset in this paper is original. However, I notice that some parts of this paper are very similar to existing benchmarking work on TAG. For example, the related work section, the dataset construction section in 4.2, the method introduction section in 4.3, and the conclusion section at the end are all similar to the [1].
Significance: The constructed TEG datasets can further enrich the study of representation learning on TAG, as well as facilitate the exploration of TEG data in the NLP and GNN communities.
  Other strengths and weaknesses can be found in the following sections.
[1] A comprehensive study on text-attributed graphs: Benchmarking and rethinking. NeurIPS2023.

**Strengths:**

1.This is an interesting work that notes the shortcomings of existing text-attributed graph dataset and constructs a new type of dataset.
2.The comprehensive textual-edge datasets constructed in this paper will foster collaboration between the NLP and GNN communities.

**Additional Feedback:**

N/A

**Clarity:**

The writing of this paper could be improved. As I read this paper, many parts of it left me confused. Here are some examples.
1. In the related work section, the authors mainly introduce the mainstream representation learning methods on TAG, but isn't this paper focused on TEG?  The authors need to show more clearly the difference and connection between the representation learning methods on TAG and TEG. Meanwhile, the description of this part of “LLM as Predictor” is not in the same style as the first two parts.
2. In the preliminaries section, there is no need to mention the challenges. These challenges have already been described in detail in the previous section.
3. On the User-Book Review Network dataset, the authors only introduce the textual information and node label information corresponding to the node when the node is a book. Do the textual information and labels exist when the node is a user? This doubt exists on both Shopping Networks and Social Networks.
4. The acquisition of edge textual information on citation networks is critical. The authors need to briefly describe the main process when constructing edge textual information with code.
5. In subsection 4.3, Eq.2 describes the learning paradigm of GNN. However, the aggregation process of traditional GNN is mainly based on node features. How the features on the edges are incorporated into the aggregation process needs further introduction by the authors.
6. For the LLM as Predictor method, the authors need to show more clearly how the prompt is constructed and what the question corresponds to on different datasets and tasks.

**Correctness:**

The TEG-DB dataset is constructed in a sound way. However, the experimental sections do not seem credible.
1. Firstly, the implementation of the PLM-based method is not carefully described in this paper. It seems to be different from the definition and implementation of PLM-based methods in previous work [1].
2. The selection of evaluation metrics is somewhat inappropriate. On the link prediction task, the paper is not evaluated on mainstream metrics such as Hits@K [2]. On the node classification task, the selection of metrics on different datasets is also not clearly explained by the authors.
3. Insufficient number of baseline models. For GNN, many classical models such as GCN, GAT, RevGAT, are not analysed experimentally, although these have been mentioned by the authors in related work. For PLM, some mainstream open-source large language models such as Llama and Mistral are not used.

[1] A comprehensive study on text-attributed graphs: Benchmarking and rethinking. NeurIPS2023.
[2]Open graph benchmark: Datasets for machine learning on graphs.NeurIPS 2020.

**Documentation:**

This paper provides links to allow the researchers with access to the dataset and provides the appropriate code to reproduce the experimental results.

**Limitations:**

The authors have adequately discussed the limitations and potential negative societal impact of their work.

**Opportunities For Improvement:**

1.Datasets need to be in a unified format. When I access the dataset links, I find that not all datasets have a unified format. The authors need to normalize the datasets and make it easy for researchers to use the individual data.
2.Additional baseline models are needed. For GNN-based method, it is necessary to add some mainstream GNNs, such as GCN, GAT, RevGAT, etc. For PLM-based method, it is recommended to add RoBERTa, Llama, Mistral, etc.
3.Need to come up with own methods. The authors performed experiments on many classical methods and obtained experimental results. Authors should further address the shortcomings of existing methods for representation learning on TEGs based on the experimental results, or propose new methods for representation learning on TEGs. This way the paper can have a more in-depth insight.
4.This paper needs to add more citations to support points. For example, in the GNN section of subsection 4.3, the author's inference that textual information on the edge cannot be captured effectively. This needs to be justified by providing more relevant citations to support the point.
5.The oobservation of the experiment do not appear to be consistent with the results of the experiment. In subsection 5.3, the authors argue that “The scale of PLMs significantly impacts the performance of TEG tasks.”. However, experimental results on node classification show that the smallest sized language model BERT performs better than the largest sized language model GPT-3.5-TURBO in most cases. The authors need to further confirm and carefully analyse the experimental results.

**Relation To Prior Work:**

Related works are addressed well.

**Summary And Contributions:**

This paper introduces Textual-Edge Graphs Datasets and Benchmark (TEG-DB). In particular, the TEG-DB datasets feature rich textual descriptions on nodes and edges and encompass a wide range of domains. Extensive experiments on TEG-DB evaluate a variety of methods. The construction of TEG-DB can advance related research in the field of Text-Attributed Graphs.

---

> ### Author Rebuttal · Authors · 2024-08-17
>
> ### Q1. Datasets need to be in a unified format.
> **All the datasets have been processed into a unified format**. The reason why you might see a non-unified format is because our dataset links also contain the raw data before processing. To avoid confusion, all the processed data has been made available as .pkl files (via the link shown in Reference[1]), ensuring consistency and ease of use. For the processed data, specifically, the `text_nodes` attribute holds node text, `text_edges` holds edge text, and `edge_index` represents the graph structure.
>
> [1] https://huggingface.co/datasets/Zixing-GOU/TEG-DB/tree/main
>
> ### Q2. Additional baseline models are needed.
> **We conducted additional experiments by adding mainstream GNNs using Bert text embedding and PLMs , including GCN, GAT, RevGAT, Llama, and Mistral. The F1 score results of the five datasets are shown in the table.** We find that (1) The performance of link prediction models varies significantly across datasets. GCN  and GAT consistently achieve high scores, especially on Reddit and Movie datasets, indicating strong generalization. RevGAT performs best on Reddit and is also effective on Amazon-Apps and Twitter, though its results on Movie and Children datasets are less impressive. In contrast, llama3 and mistral generally perform poorly across all datasets except Twitter, particularly on Reddit and Amazon-Apps, suggesting they are less effective for Textual-Edge Graphs. (2) For node classification, GCN and GAT exhibit strong performance across most datasets, especially on Reddit and Amazon-Movie, while RevGAT also performs well but with a slight edge on Reddit. llama3 shows mixed results with decent performance on Amazon-Apps and Reddit, but poor results on Amazon-movie and Twitter, while mistral generally underperforms across all datasets except Reddit.
>
> | Link Prediction | Reddit | Amazon-Apps  | Twitter | Amazon-Movie  | Children  |
> |-----------------|--------|--------------|---------|--------|-----------|
> | GCN             | 0.9720 | 0.7798       | 0.5000  | 0.7868 | 0.8619    |
> | GAT             | 0.9546 | 0.7455       | 0.5000  | 0.7881 | 0.8398    |
> | RevGAT          | 0.9801 | 0.7422       | 0.5945  | 0.7882 | 0.8398    |
> | llama3          | 0.7561 | 0.4118       | 0.5816  | 0.1955 | 0.6626    |
> | mistral         | 0.8463 | 0.2803       | 0.7092  | 0.1636 | 0.684     |
>
>
> | Node Classification | Reddit | Amazon-Apps  | Twitter | Amazon-Movie  | Children  |
> |---------------------|--------|--------------|---------|--------|-----------|
> | GCN                 | 0.9532 | 0.1625       | 0.8144  | 0.9962 | 0.7024    |
> | GAT                 | 0.9548 | 0.2047       | 0.8141  | 0.9962 | 0.7263    |
> | RevGAT              | 0.9723 | 0.1514       | 0.8146  | 0.9962 | 0.7263    |
> | llama3              | 0.8406 | 0.4118       | /       | 0.3865 | 0.5179    |
> | mistral             | 0.2065 | 0.2803       | /       | 0.6368 | 0.4911    |

---

> > ### Author Rebuttal · Authors · 2024-08-17
> >
> > ### Q3. Need to come up with own methods.
> > 1. We have come up with our own methods for TEGs. As TEGs are an area not well explored yet,  GNNs are typically designed for structured attributed edges, and hence this is why **we extended them by adding a text embedding module to first process the edge texts into edge embedding to be used by GNNs thereafter**. Such methods are named GraphSAGE, GINE, EdgeConv, GeneralConv, and GraphTransformer in our paper.
> > 2. We further introduced our own methods for TEGs. Traditional approaches that first learn edge text embeddings and then apply GNNs have limitations for TEG data because edge texts and node texts are often closely entangled. Separating them into distinct node and edge embeddings may impair important information regarding their interaction. For instance, in a citation graph where each node represents a paper, an edge might indicate that one paper cites, criticizes, or utilizes a specific part of another paper. Therefore, the edge does not represent the relationship between the entirety of the two nodes, posing a significant challenge for methods that rely on node or edge embeddings representing the entirety of a node or edge. **To avoid information loss during the interaction between nodes and edges after text embedding, we propose an approach that first entangles the edge text and node text before performing the embedding.** The embedding obtained in this way is then added to the message-passing operation for each pair of connected nodes. The advantage of this method over existing approaches is its ability to effectively preserve the semantic relationships between nodes and edges, making it more suitable for capturing complex relationships. **The F1 score results of  the five datasets with Bert text embedding are shown as follows:**
> >
> >
> > |                   | Reddit               |                     | Amazon-Apps          |                     | Twitter              |                     | Amazon-Movie         |                     | Children             |                      |
> > |-------------------|----------------------|---------------------|----------------------|---------------------|----------------------|---------------------|----------------------|---------------------|----------------------|----------------------|
> > | Link Prediction   | Entangled Embeddings | Seperate Embeedings | Entangled Embeddings | Seperate Embeedings | Entangled Embeddings | Seperate Embeedings | Entangled Embeddings | Seperate Embeedings | Entangled Embeddings | Seperate Embeedings  |
> > | GINE              | 0.9975               | 0.9808              | 0.8601               | 0.8123              | 0.7962               | 0.7880              | 0.8921               | 0.8456              | 0.9049               | 0.8914               |
> > | GraphTransformer  | 0.9980                | 0.9808              | 0.8122               | 0.7846              | 0.8386               | 0.8165              | 0.8648               | 0.8185              | 0.8804               | 0.8763               |
> > | GCN               | 0.9823               | 0.9720              | 0.8437               | 0.7798              | 0.5206               | 0.5000              | 0.8359               | 0.7868              | 0.8691               | 0.8619               |
> > | GAT               | 0.9742               | 0.9546              | 0.8732               | 0.7455              | 0.5126               | 0.5000              | 0.8365               | 0.7881              | 0.8399               | 0.8398               |
> > | RevGAT            | 0.9857               | 0.9601              | 0.8681               | 0.7422              | 0.6094               | 0.5945              | 0.8393               | 0.7882              | 0.8399               | 0.8398               |
> >
> >
> > |                     | Reddit               |                     | Amazon-Apps          |                     | Twitter              |                     | Amazon-Movie         |                     | Children             |                      |
> > |---------------------|----------------------|---------------------|----------------------|---------------------|----------------------|---------------------|----------------------|---------------------|----------------------|----------------------|
> > | Node Classification | Entangled Embeddings | Seperate Embeedings | Entangled Embeddings | Seperate Embeedings | Entangled Embeddings | Seperate Embeedings | Entangled Embeddings | Seperate Embeedings | Entangled Embeddings | Seperate Embeedings  |
> > | GINE                | 0.9972               | 0.9965              | 0.4592               | 0.3592              | 0.8549               | 0.8328              | 0.5383               | 0.5085              | 0.7598               | 0.7552               |
> > | GraphTransformer    | 0.9970                | 0.9966              | 0.4918               | 0.3818              | 0.8356               | 0.8261              | 0.5543               | 0.5124              | 0.7586               | 0.7491               |
> > | GCN                 | 0.9863               | 0.9532              | 0.2525               | 0.1625              | 0.8145               | 0.8144              | 0.5017               | 0.4756              | 0.7354               | 0.7263               |
> > | GAT                 | 0.9533               | 0.9548              | 0.3047               | 0.2047              | 0.8145               | 0.8141              | 0.5206               | 0.4795              | 0.7263               | 0.7235               |
> > | RevGAT              | 0.9869               | 0.9723              | 0.2614               | 0.1514              | 0.8156               | 0.8146              | 0.5382               | 0.4961              | 0.7265               | 0.7231               |

---

> > ### Author Rebuttal · Authors · 2024-08-17
> >
> > We can find that (1) For link prediction, GraphTransformer and GINE demonstrate a notable advantage when using entangled embeddings, consistently achieving the highest performance across most datasets; in contrast, models using separate embeddings generally show lower performance, highlighting the effectiveness of entangled embeddings in capturing complex semantic relationships in the data. (2) For node classification, entangled embeddings provide a clear advantage, as evidenced by GINE and GraphTransformer achieving significantly higher performance with entangled embeddings across most datasets. Specifically, these models outperform their counterparts using separate embeddings, especially on Reddit and Twitter, where the gains are most pronounced. In contrast, the models using separate embeddings generally show lower accuracy, underscoring the effectiveness of entangled embeddings in capturing complex relationships within the data.

---

> > ### Author Rebuttal · Authors · 2024-08-25
> >
> > ### Q4. This paper needs to add more citations to support points.
> > 1. Existing Graph ML methods like GNNs typically work on structured attributes on edges instead of texts. In TEGs, edges are texts that contain rich semantic information, which is way beyond the GNNs that are commonly based on connectivity (i.e., binary attribute denoting whether there is a connection or not) and edge attributes (i.e., categorical or numerical values on the edges) [1][2].
> >
> > 2. Transforming edge texts into edge embedding still has its limitations. Learning edge text embedding and then applying GNN has its limitation for TEGs data due to its unique characteristics because edge texts and node texts can be well entangled and hence modulizing them into node embedding and edge embedding may impair important information of their entanglement [3]. For example, in a citation graph where each node is a paper and an edge can mention one paper is citing/criticizing/utilizing a specific part in another paper. Hence, this edge does not denote the relationship between the entirety of one node and the entirety of another, and hence seriously challenging the methods based on the calculating of node/edge embedding, which is a representation of the entirety of a node/edge.
> >
> > [1] Jin, Bowen, et al. "Large language models on graphs: A comprehensive survey." arXiv preprint arXiv:2312.02783 (2023).
> >
> > [2] Fan, Wenqi, et al. "Graph machine learning in the era of large language models (llms)." arXiv preprint arXiv:2404.14928 (2024).
> >
> > [3]  Yan, Hao, et al. "A comprehensive study on text-attributed graphs: Benchmarking and rethinking." Nips 23
> >
> > ### Q5. The observations of the experiment do not appear to be consistent with the results of the experiment.
> > Thank you for your comments and suggestions. We realize that the statement of our observation analysis may need to be better elaborated in more detail. In this paper,  the methods we experiment with are primarily based on GNN and PLM.
> >
> > 1. For the GNN methods use GNN as the predictor. Specifically, the scale of PLMs significantly impacts the performance of TEG tasks when text is sufficiently rich in nodes and edges (e.g., Goodreads-Children dataset with 350 average edge text length). This is because more expressive large models have bigger power in handling more sophisticated rich semantics and thereby generate superior embeddings by outperforming other methods by 2% on average in node classification.  Otherwise, when texts are relatively fewer in nodes and edges,  (e.g., Amazon-Apps dataset with 50 average edge text length), methods with large language models may not necessarily outperform those with smaller language models because both large and small language models might adequately capture the available semantics with limited text to interpret.
> > 2. For the PLM methods use MLP as the predictor. Specifically, these methods using MLP as the predictor appear to not be significantly affected by the size of the PLMs when generating the embeddings,  largely because the simplicity of the MLP may bottleneck the full utilization of the PLM embeddings.
> >
> > ### Q6. The implementation of the PLM-based method is not carefully described in this paper.
> > Thank you for pointing this out. TEGs are different from traditional data and hence needs some adaptation of existing methods. Specifically, the difference between our PLM-based method and previous methods is that we have added an additional MLP in order to incorporate edge information from TEGs.

---

> > ### Author Rebuttal · Authors · 2024-08-25
> >
> > ### Q7. The selection of evaluation metrics is somewhat inappropriate.
> > **Link Prediction:**  The use of AUC and F1-score as evaluation metrics for link prediction tasks is widely accepted in the literature, as demonstrated by works such as ogbl-vessel [1] and DTGB [2]. We acknowledge the reviewer's suggestions and further expand our evaluation metrics to include Hits@50 on the reddit dataset link prediction using Bert text embedding. This addition will offer a more comprehensive evaluation of our method. We plan to extend these metrics to all datasets in future work.
> >
> > |Reddit link prediction|Hits@50|
> > |:---:|:---:|
> > |GCN|0.9200|
> > |GAT|0.9316|
> > |RevGAT|0.9428|
> > |llama3|0.6200|
> > |mistral|0.6312|
> >
> > As shown in the table, the method of revgat is the best while llama3 is the worst.
> >
> > **Node classification:** As mentioned in our paper, the nature of the datasets varies, leading to some tasks being multi-label classification and others being multi-class classification. For multi-label classification, we use AUC-micro and F1-micro as evaluation metrics, while for multi-class classification, we employ Accuracy (ACC) and F1-score. These metrics are chosen to reflect the performance characteristics of each classification type appropriately.
> >
> > [1] Hu, Weihua, et al. "Open graph benchmark: Datasets for machine learning on graphs." Advances in neural information processing systems 33 (2020): 22118-22133.
> >
> > [2] Zhang, Jiasheng, et al. "DTGB: A Comprehensive Benchmark for Dynamic Text-Attributed Graphs." arXiv preprint arXiv:2406.12072 (2024).
> >
> > ### Q8.  Difference and connection between TAG and TEG
> > 1. The reason we mentioned a lot about TAG in related work instead of TEG is because TEG is a highly under-explored area with barely any existing benchmarks, and TEG is an extension derived from TAG so the introduction of TAG provides important background.
> > 2. Difference between TAG and TEG. TAGs are defined as “graphs where nodes are often associated with text attributes” [1, 2, 3], while TEGs require that edges are texts, as defined in our paper.
> >
> > [1] Yan, Hao, Chaozhuo Li, Ruosong Long, Chao Yan, Jianan Zhao, Wenwen Zhuang, Jun Yin et al. "A comprehensive study on text-attributed graphs: Benchmarking and rethinking." Advances in Neural Information Processing Systems 36 (2023): 17238-17264.
> > [2] Yang, Junhan, Zheng Liu, Shitao Xiao, Chaozhuo Li, Defu Lian, Sanjay Agrawal, Amit Singh, Guangzhong Sun, and Xing Xie. "Graphformers: Gnn-nested transformers for representation learning on textual graph." Advances in Neural Information Processing Systems 34 (2021): 28798-28810.
> > [3] He, Xiaoxin, Xavier Bresson, Thomas Laurent, Adam Perold, Yann LeCun, and Bryan Hooi. "Harnessing explanations: Llm-to-lm interpreter for enhanced text-attributed graph representation learning." arXiv preprint arXiv:2305.19523 (2023).
> >
> > ### Q9. In the preliminaries section, there is no need to mention the challenges
> >
> > We mentioned the challenges at the end of the preliminaries section as we intended to use these challenges to motivate the methods introduced in the technique section. We will take your advice to simplify the challenges and reduce the duplication.
> >
> > ### Q10.  Textual and labels in datasets
> > In User-Book Review, Shopping, and Social Networks, user node text is the user’s name and information, but there are no associated labels. Therefore, in node classification, we classify other nodes, but not user nodes.
> >
> > ### Q11. Edge textual information with code
> > In citation networks, edge text represents the citation context between connected papers. We use the Semantic Scholar API to retrieve and associate this citation data with the network's edges.
> >
> > ```python
> > url = f"https://api.semanticscholar.org/graph/v1/paper/{paper_id}/citations"
> > response = requests.get(url, headers=headers)
> > ```
> > ### Q12.  Edge features into aggregation process
> > Most GNN baselines, like GeneralConv, GINE, and GraphTransformer, handle edge features. For those that don't, we linearly map edge attributes and incorporate them into node representations during message passing.

---

> > > ### Author Rebuttal · Authors · 2024-08-25
> > >
> > > ### Q13. For the LLM as Predictor method, the authors need to show more clearly how the prompt is constructed and what the question corresponds to on different datasets and tasks.
> > >
> > > Below are the prompts we used for the Amazon-app and Reddit datasets for the link prediction and node classification tasks:
> > >
> > > **Amazon-apps link prediction**
> > > ```
> > > Please predict whether there should be a link between two nodes based on the
> > > Amazon-apps dataset, where nodes represent reviewers and items  and edges represent reviews. Below is a list of
> > > node pairs with their descriptions.
> > > [Node Pairs]
> > >
> > > Your task is to predict the presence of a link
> > > between these node pairs using the following rules:
> > > 0: No link 1: Has link
> > > Your response should include only one relevant code and its corresponding
> > > label name, separated by a comma. Here’s the required format:
> > > Example Response:
> > > 0, no link 1, has link
> > > Now, please provide the prediction using this format.
> > > ```
> > >
> > > **Amazon-apps node classification**
> > > ```
> > > Please classify an item from the Amazon-apps dataset, where nodes represent items and edges represent reviews.
> > > Below is a list of Amazon-apps label classifications along with a sample context
> > > describing the node and the edges linked to it.
> > > [Node Pairs]
> > >
> > > Based on the provided context,
> > > determine the classification code(s) for the node.
> > > 1)Your response should include the relevant classification code(s), separated
> > > by commas.
> > > 2)No additional text should be included in your reply.
> > > 3)Each item can have multiple classifications.
> > > Example Response: 0,6 3,19,2,5
> > > Now, provide your classification using this format.
> > > ```
> > > **Reddit node classification**
> > > ```
> > > Predict whether there should be a link between two nodes based on the Reddit
> > > dataset, where nodes represent users and topics and edges represent reviews. Below is a list of node pairs
> > > with their descriptions.
> > > [Node Paris]
> > >
> > > Your task is to determine the presence of a link between
> > > these node pairs using the following rules:
> > > 0: No link 1: Has link
> > > Your response should include only one relevant code and its corresponding
> > > label name, separated by a comma. Use the format provided below:
> > > Example Response:
> > > 0, no link 1, has link
> > > Now, please provide your prediction using this format.
> > > ```
> > > **Reddit node classification**
> > > ```
> > > The Reddit dataset contains nodes representing users and 204 topics. Your task
> > > is to classify these nodes based on the context provided, where nodes represent users and topics and edges represent reviews.
> > > [Node pairs]
> > >
> > > Below is a list of Reddit
> > > label classifications, each indicating the category to which a node belongs, along
> > > with a sample context describing the node and its linked edges to help you make
> > > your judgment.
> > > 1)Based on the context, determine the label classification code for the node.
> > > 2)Your response should include only one relevant code and its corresponding
> > > label name, separated by a comma.
> > > 3)Only provide the most relevant code that you believe applies.
> > > Example Response:
> > > 0, user nodes, not a moderator for Reddit community
> > > Now, provide your classification using this format.
> > > ```

---

> ### Comment · Reviewer_r2GB · 2024-08-26
> **Response to the rebuttal of the reviewers**
>
> Thank you for the reviewer rebuttal. Some of my previous concerns are addressed. I would like to raise my score to 6: Marginally above acceptance threshold.

---

### Decision · Program_Chairs · 2024-09-26

**Decision:**

Accept (Poster)

**Comment:**

This paper focuses on text-attributed graphs and introduces the first textual-edge graphs datasets and benchmarks (TEG-DB), a comprehensive and diverse collection of benchmark textual-edge datasets. Additionally, they conduct extensive benchmark experiments on TEG-DB to evaluate the current progress. The constructed graph might facilitate new research on textual-edge graphs. In general, the construction of the dataset is good, but both writing and experimental parts of the paper should be improved. Additional baseline models are needed. The authors addressed most of the issues mentioned above during rebuttal. I recommend acceptance of the paper. If accepted, the authors should incorporate some of the responses and experimental results during rebuttal to the camera-ready version.